# REMOTE SENSING-ORIENTED WORLD MODEL

## ABSTRACT

World models have shown potential in artificial intelligence by predicting and reasoning about world states beyond direct observations. However, existing approaches are predominantly evaluated in synthetic environments or constrained scene settings, limiting their validation in real-world contexts with broad spatial coverage and complex semantics. Meanwhile, remote sensing applications urgently require spatial reasoning capabilities for disaster response and urban planning. This paper bridges these gaps by introducing the first framework for world modeling in remote sensing. We formulate remote sensing world modeling as direction-conditioned spatial extrapolation, where models generate semantically consistent adjacent image tiles given a central observation and directional instruction. To enable rigorous evaluation, we develop RSWISE (Remote Sensing World-Image Spatial Evaluation), a benchmark containing 1,600 evaluation tasks across four scenarios: general, flood, urban, and rural. RSWISE combines visual fidelity assessment with instruction compliance evaluation using GPT-4o as a semantic judge, ensuring models genuinely perform spatial reasoning rather than simple replication. Afterwards, we present RemoteBAGEL, a unified multimodal model fine-tuned on remote sensing data for spatial extrapolation tasks. Extensive experiments demonstrate that RemoteBAGEL consistently outperforms state-of-the-art baselines on RSWISE.

## 1 INTRODUCTION

World models have emerged as a frontier in artificial intelligence, showing promise across diverse applications such as robotic navigation (Wu et al., 2023) and autonomous driving (Guan et al., 2025). These models aim to learn the compressed latent representations of environments from limited observations and to predict or reason about unobserved states by simulating the underlying dynamics in this latent space (Ding et al., 2025). However, most world model studies remain confined to synthetic simulators or constrained scene settings. Synthetic settings lack the complexity and uncertainty of real environments, while constrained scene settings fail to capture reasoning over large spatial structures. As a result, the real-world effectiveness of current world models in spatial reasoning remains largely untested.

Remote sensing provides a uniquely powerful testbed for world models. Satellite and aerial imagery naturally encode "world-level" structures such as urban road networks (Yu & Fang, 2023), river systems (Tomsett & Leyland, 2019), agricultural mosaics (Khanal et al., 2020), and forest landscapes (Fassnacht et al., 2024). At the same time, high-impact applications—including flood prediction for disaster response (Nguyen et al., 2024) and infrastructure forecasting in urban planning (Wellmann et al., 2020)—require reasoning beyond directly observed regions. Yet, much of remote sensing research has focused on recognition tasks such as classification (Li et al., 2022; Temenos et al., 2023) and semantic segmentation (Sun et al., 2020; Zhang et al., 2023), leaving the potential of world modeling in this domain unexplored.

This paper bridges these gaps by introducing the first framework for world modeling in remote sensing. We formulate remote sensing world modeling as direction-conditioned spatial extrapolation (defined in the image-grid frame with up, down, left, and right, rather than geographic cardinal directions), where models generate semantically consistent adjacent image tiles given a central observation and directional instruction. As illustrated in Figure 1, this formulation explicitly models the spatial transition as a next-state prediction task, requiring inference over the unobserved world structure.

Figure 1: Illustration of direction-conditioned spatial extrapolation for Remote Sensing World Modeling. Given a central observation $x_t$ (the current spatial tile) and a directional instruction $a_t$, the world model learns the underlying geospatial structure and predicts the adjacent, previously unobserved tile $x_{t+1}$. Here, the index $t$ denotes spatial progression rather than temporal evolution, aligning our task with the next-state prediction paradigm used in World Models.

However, introducing world models to the remote sensing domain faces a fundamental evaluation challenge rooted in the limitations of existing assessment paradigms. Current evaluation approaches suffer from critical methodological flaws across two distinct failure modes. Distributional fidelity metrics such as Fréchet Inception Distance (FID) (Heusel et al., 2017) measure statistical realism but ignore whether generated tiles follow spatial instructions. As a result, models could obtain deceptively low FID scores by replicating inputs or producing visually plausible but spatially incoherent imagery. Conversely, large language model–based semantic evaluations such as World Knowledge-Informed Semantic Evaluation (WISE) (Niu et al., 2025) capture instruction-following semantics but lack quantitative grounding in distributional realism, making it difficult to detect fine-grained issues such as spatial discontinuities or texture mismatches.

To overcome these limitations, we introduce RSWISE (Remote Sensing World-Image Spatial Evaluation), the first evaluation framework designed for remote sensing world models. RSWISE integrates distributional fidelity with spatial reasoning consistency through a dual-dimension approach. Specifically, it employs FID to ensure adherence to real-world satellite statistics and leverages GPT-4o to assess whether generated tiles reveal novel yet geographically plausible regions aligned with directional prompts. As shown in Figure 2, RSWISE reflects better geospatial alignment than FID alone, providing a principled basis for fair comparison and progress tracking.

Building on BAGEL (Deng et al., 2025), a unified multimodal model for generation and understanding, we introduce RemoteBAGEL, the first remote sensing world model specifically designed for direction-conditioned spatial extrapolation. In contrast to prior methods that may yield visually plausible yet spatially inconsistent completions, RemoteBAGEL explicitly couples the generative process with spatial reasoning constraints. It is built around two components: (1) a trajectory-based data construction pipeline that transforms raw satellite imagery into instruction-conditioned continuation tasks, and (2) a reconstruction-driven training framework that enforces geographic continuity and semantic coherence during spatial extrapolation. Extensive experiments demonstrate that RemoteBAGEL consistently outperforms state-of-the-art baselines on RSWISE. In summary, our contributions are threefold:

- We propose a novel problem formulation for remote sensing world modeling as direction-conditioned spatial extrapolation tasks;

- We introduce RSWISE, the first comprehensive evaluation framework with dual-dimension metrics and a benchmark of 1,600 evaluation tasks across four representative scenarios;

- We develop RemoteBAGEL, the first specialized world model achieving state-of-the-art performance in remote sensing spatial reasoning tasks.

| Benchmark | #Examples | Application Context | Evaluation Metric | Scen. Div. | Geo. Sem. |
|---|---|---|---|---|---|
| VBench Ji et al. (2024) | 800 | Video | FID / Human | ✗ | ✗ |
| WorldModelBench Li et al. (2025) | 350 | Games / Video | Task-specific | ✓ | ✗ |
| ChronoMagic-Bench Yuan et al. (2024) | 1649 | Video | Temporal Consistency | ✗ | ✗ |
| TC-Bench Feng et al. (2024) | 150 | Video | FID | ✗ | ✗ |
| RSWISE (Ours) | 1600 | Remote Sensing | FID + GPT semantic | ✓ | ✓ |

Table 1: Comparison of benchmarks. Column headers are abbreviated for readability: Scen. Div. = *Scenario Diversity*, and Geo. Sem. = *Geospatial Semantics*. Existing benchmarks mainly evaluate temporal prediction in robotics, video, or game settings. In contrast, RSWISE (Remote Sensing World-Image Spatial Evaluation) provides 1,600 evaluation tasks, constructed from 100 images × 4 scenarios × 4 directions. It focuses on spatial continuation in remote sensing, leveraging real geospatial imagery and explicitly evaluating semantic continuity of structures such as rivers, roads, and urban–rural transitions.

## 2 RELATED WORK

### 2.1 WORLD MODELS AND BENCHMARKS

World models generally divided into two perspectives: models that aim to understand the world by abstracting its underlying mechanisms, and models that aim to predict the future by simulating possible evolutions of the environment (Ding et al., 2025). Early efforts such as *World Models* (Ha & Schmidhuber, 2018) focused on abstracting the external world to gain a deep understanding of its underlying mechanisms, while subsequent work including PlaNet (Hafner et al., 2019) and the Dreamer family (Hafner et al., 2020; 2022) introduced recurrent state-space models (RSSMs) that designed to facilitate forward prediction purely within the latent space. More recent advances extend this principle into generative modeling: transformer-based models such as TransDreamer (Chen et al., 2024) and Genie (Bruce et al., 2024), diffusion and VAE-driven approaches for scene extrapolation and controllable driving (Wang et al., 2025; Cai et al., 2023), and the JEPA family (Assran et al., 2023) that reframe world models as self-supervised abstraction learners.

Beyond these task-specific designs, emerging unified multimodal models (UMMs) such as BAGEL (Deng et al., 2025) and Qwen-Image-Edit (Wu et al., 2025) demonstrate the capacity to jointly support both perception (understanding the input image and directional instruction) and generation (synthesizing the new tile). This inherent capability for unified understanding and generation, which includes learning compressed representations of the input and inferring and generating new states, positions UMMs as potential foundation world models. They naturally handle the multimodal input and the complex generation task, overcoming the need for separate, modular components. This work explores how such unified architectures perform when extended to spatial reasoning and world modeling tasks in the remote sensing domain.

Despite this diversity, evaluation remains a central challenge: existing benchmarks such as VBench (Ji et al., 2024), ChronoMagic-Bench (Yuan et al., 2024), TC-Bench (Feng et al., 2024), and WorldModelBench (Li et al., 2025) focus on controlled or synthetic scene settings, but they do not explicitly involve spatial continuity or geospatial semantics at the remote sensing scale in real-world contexts. This gap prevents current evaluations from testing how well world models reason over real-world structures such as rivers, roads, or urban–rural transitions, as summarized in Table 1.

### 2.2 REMOTE SENSING MODELS

Remote sensing (RS) acquires Earth observations from satellites and aerial platforms, producing imagery that encodes both spectral variation and large-scale spatial structures across diverse environments (Li et al., 2019). Much of RS research has traditionally focused on recognition-oriented tasks, including land-cover classification and semantic segmentation (Sun et al., 2020; Zhang et al., 2023). While recent advances have led to large-scale RS foundation models such as SkySense (Guo et al., 2024) and SpectralGPT (Hong et al., 2024) that excel at learning transferable representations, these methods seldom attempt spatial continuation or reasoning over large geospatial structures. However, generative AI is an emerging field in RS, with diffusion models being applied to tasks such as super-resolution, cloud removal, and metadata-conditioned image synthesis (Huang et al., 2025).

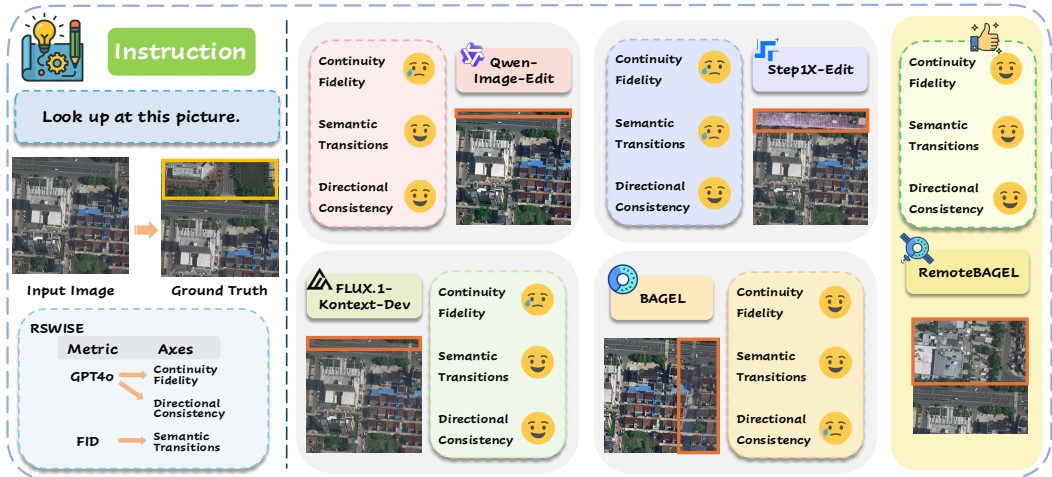

Figure 2: Overview of the RSWISE evaluation framework. RSWISE assesses spatial extrapolation quality along three axes-*continuity fidelity*, *semantic transitions*, and *directional consistency*. These axes are jointly operationalized through two complementary metrics: GPT-4o for spatial reasoning and FID for distributional fidelity. The five examples illustrate how models differ across these axes: some achieve low FID yet fail to produce meaningful directional content, while others show strong spatial reasoning and continuity but incorrect directional consistency. RSWISE integrates both aspects to provide a balanced and geospatially grounded assessment of world modeling performance.

Across these generative methods-including Text2Earth (Liu et al., 2025a) and EMRDM (Liu et al., 2025c)-the focus remains strictly on within-tile image synthesis or restoration, rather than modeling the latent structure of the Earth surface or predicting unobserved world states. Therefore, we explore world modeling as a new paradigm for RS, introducing RemoteBAGEL to explicitly couple unified multimodal generation with the spatial reasoning requirements inherent to Earth observation data.

## 3 THE RSWISE BENCHMARK

**Design overview.** The goal of RSWISE is to provide a comprehensive evaluation framework for remote sensing world models that directly addresses the challenge of spatial reasoning in geospatial contexts. It is built around three components: (1) a unified formulation of directional spatial extrapolation, (2) a multi-scenario dataset capturing diverse conditions, and (3) dual-dimension metrics jointly assessing fidelity and reasoning.

### 3.1 SPATIAL CONTINUATION SPECIFICATION

**Problem formulation.** We formalize remote sensing world modeling as a directional spatial extrapolation task. Each evaluation instance is represented by a triplet $(T_{\text{input}}, I_{\text{dir}}, T_{\text{target}})$, where $T_{\text{input}}$ denotes the observed central tile of a geographic region, $I_{\text{dir}}$ is the directional instruction, and $T_{\text{target}}$ is the ground-truth adjacent tile. The objective is to model the conditional distribution

$$p_\theta(T_{\text{target}} \mid T_{\text{input}}, I_{\text{dir}}), \tag{1}$$

and generate a tile at inference time via

$$T_{\text{generated}} \sim p_\theta(\cdot \mid T_{\text{input}}, I_{\text{dir}}). \tag{2}$$

The requirement is that $T_{\text{generated}}$ achieves distributional fidelity with real satellite imagery while preserving semantic coherence with the specified spatial direction. Importantly, directions are defined in the image-grid coordinate frame (up, down, left, right) rather than cardinal North–South–East–West; our analyses therefore study anisotropy in grid-aligned continuations independent of geographic orientation.

**Evaluation axes.** To capture the complexity of spatial extrapolation, RSWISE defines three complementary axes: (1) *continuity fidelity*, requiring generated tiles to extend geographic structures across boundaries (e.g., roads, rivers, vegetation patches); (2) *semantic transitions*, requiring plausible changes across heterogeneous regions (e.g., urban to rural, land to water); and (3) *directional consistency*, requiring strict adherence to the instructed direction. These axes ensure that evaluation moves beyond visual plausibility and directly probes spatial reasoning.

## 3.2 DATASET CURATION

To comprehensively evaluate spatial extrapolation under diverse geospatial conditions, we define four representative scenarios that capture complementary challenges in remote sensing world modeling. The first is the general setting of geographic extrapolation, designed to evaluate model capability across diverse scene types rather than within a single environment, thereby enabling a more comprehensive assessment. Beyond this general setting, flood scenarios introduce highly dynamic environmental variations, urban regions emphasize continuity across dense built environments, and rural landscapes highlight consistency within natural and agricultural patterns. These scenarios jointly establish a structured basis for assessing spatial extrapolation across stable, dynamic, structured, and natural contexts.

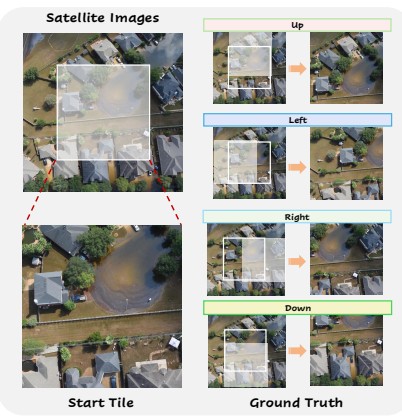

Figure 3: Start tile is paired with their four cardinal neighbors, yielding evaluation triplets.

The benchmark dataset consists of 1,600 curated evaluation instances evenly distributed across these four scenarios: general, flood, urban, and rural. The data are sourced from three publicly available datasets: Sky-SA (Zhu et al., 2025) for general scenes, FloodNet (Rahnemoonfar et al., 2021) for flood events, and LoveDA (Wang et al., 2021) for urban and rural landscapes.

Although built from three public datasets, the resulting benchmark exhibits substantial diversity: it spans imagery collected across multiple cities and geographically distinct regions, covers a wide range of spatial resolutions from ultra–high-resolution UAV imagery ($\sim$1.5 cm GSD) to satellite imagery at 0.3 m GSD, and includes more than 1,700 distinct semantic categories represented across varied land-cover types. These characteristics ensure that RSWISE evaluates spatial extrapolation under diverse geospatial structures rather than a narrow or dataset-specific distribution.

**Geospatial scenario taxonomy.**

- *General*: diverse landscapes including mountains, forests, coastlines, and mixed terrain, serving as a baseline across varied topographies.
- *Flood*: disaster-response contexts with inundated areas and disrupted land cover, testing robustness under dynamic environmental perturbations.
- *Urban*: dense built environments with road networks and building layouts, challenging models to reason over structured spatial patterns.
- *Rural*: agricultural regions, natural vegetation, and sparse settlements, emphasizing the continuity of natural patterns and land-use transitions.

**Data construction pipeline.** For each scenario, large satellite images are divided into $3 \times 3$ overlapping grids of tiles, where each tile overlaps with its neighbors by approximately 66.7% along both horizontal and vertical directions, ensuring boundary consistency and preserving spatial autocorrelation. Start tiles are paired with their four cardinal neighbors (up, down, left, right), yielding evaluation triplets. Directional instructions are standardized into fixed prompts (e.g., "Look at what is below this picture") to ensure fairness across models. Filtering criteria include cloud cover thresholds, resolution consistency, and temporal alignment. A quality assurance process-combining automated

checks for artifacts, manual inspection of geographic coherence, and balanced sampling—produces 400 instances per category.

### 3.3 RSWISE EVALUATION METRICS

The three evaluation axes-continuity fidelity, semantic transitions, and directional consistency-specify the conceptual dimensions along which spatial extrapolation should be assessed. Consequently, these axes are operationalized in RSWISE via two complementary metrics: *distributional fidelity*, which measures the alignment of generated tiles with the statistical properties of real satellite imagery, and *spatial reasoning*, which assesses whether generated tiles follow the instructed direction of extrapolation. These metrics provide a concrete instantiation of the conceptual framework, ensuring that both realism and reasoning are jointly evaluated.

**Distributional Fidelity.**   We employ FID to quantify how well generated tiles align with the statistical properties of real satellite imagery. For remote sensing applications, FID reflects whether generated content follows scenario-specific geographic distributions such as urban density, vegetation cover, or terrain patterns. To facilitate composite scoring, FID values are globally normalized into $s_{\text{fid}} \in [0, 1]$, standardizing units across scenarios and inverting the metric so that higher values correspond to better performance. This transformation ensures comparability across contexts and allows seamless integration with reasoning-based scores.

**Spatial Reasoning.**   We leverage GPT-4o as an external evaluator to assess whether generated tiles reflect meaningful extrapolation in the instructed direction. Valid outputs include feature continuations (e.g., rivers, roads, mountain ridges), coherent land-use transitions (e.g., urban to suburban, forest to agricultural), and natural boundary progressions (e.g., coastlines, watershed divides). The evaluator assigns scores on a $[0, 10]$ scale based on spatial coherence, directional accuracy, and geographic plausibility, which are then normalized to $s_{\text{spatial}} \in [0, 1]$. This metric explicitly penalizes models that replicate the input texture without introducing new, directionally consistent content.

**RSWISE.**   The final score integrates both fidelity and reasoning via a weighted sum:

$$\text{RSWISE}(m, s) = 100 \cdot \Big( w_{\text{spatial}} \cdot s_{\text{spatial}}(m, s) + w_{\text{fid}} \cdot s_{\text{fid}}(m, s) \Big), \tag{3}$$

where $m$ denotes the model and $s$ the scenario. We assign greater weight to spatial reasoning while retaining distributional fidelity as a grounding constraint. A representative setting is adopted as the default for RSWISE. For a detailed validation and sensitivity analysis of the weighting scheme, please refer to Appendix C.

## 4   REMOTE SENSING-ORIENTED WORLD MODEL

We present **RemoteBAGEL**, a remote sensing world model that performs direction-conditioned spatial extrapolation via action-conditioned tile completion. Our approach has two components: (1) a trajectory-based data construction pipeline that converts unlabeled satellite imagery into action-conditioned continuation tasks, and (2) a reconstruction-centric training objective and architecture that enable controllable spatial extrapolation. We first describe the action-conditioned formulation, then detail the training methodology and the architecture overview.

### 4.1   ARCHITECTURE OVERVIEW

As illustrated in Figure 4 (c), our architecture employs a unified generative framework where the input tile undergoes feature extraction through a visual encoder, while the directional action is transformed into a learned embedding space. These representations are subsequently integrated via cross-modal and self-attention mechanisms to capture spatial-semantic dependencies. The fused features are then processed by a generative decoder to synthesize the geographically adjacent tile in the specified direction. This conditioning paradigm enables precise directional control over spatial extrapolation while preserving the structural and semantic coherence inherent in the source imagery.

Figure 4: Illustration of the RemoteBAGEL formulation. (a) Large satellite images are partitioned into overlapping $3 \times 3$ grids, with example trajectories providing consecutive steps of supervision. (b) Given a central tile and a directional instruction (up, down, left, right), the adjacent tile in the specified direction serves as ground truth, yielding instruction-conditioned triplets. (c) The architecture encodes the input tile and instruction embedding, fuses them via attention, and decodes a continuation tile consistent with the specified direction.

## 4.2 ACTION-CONDITIONED DATA CONSTRUCTION

We construct supervision directly from raw satellite imagery without human annotation. As illustrated in Figure 4 (a), large images $X \in \mathbb{R}^{H \times W \times 3}$ are partitioned into overlapping $3 \times 3$ grids $\{x_i\}_{i=1}^9$, where overlaps preserve boundary consistency and capture the spatial autocorrelation characteristic of geospatial data. For each central tile $x_c$, we define a discrete action $a \in \{\text{up}, \text{down}, \text{left}, \text{right}\}$ that specifies a directional move. This yields training triplets

$$(x_c, a, x_{\text{target}}), \quad x_{\text{target}} = \text{adjacent}(x_c, a). \tag{4}$$

in which the adjacent tile in the instructed direction serves as ground truth (Figure 4 (b)). Trajectories (an example route is shown in Figure 4 (a)) provide consecutive steps of supervision, naturally enforcing spatial continuity across tile transitions.

## 4.3 ACTION-CONDITIONED TRAINING

Given $(x_c, a, x_{\text{target}})$, the model learns a direction-conditioned completion mapping:

$$f_\theta(x_c, a) \; \rightarrow \; \hat{x}_{\text{target}} \tag{5}$$

and is trained with a reconstruction objective between the prediction and the true neighbor:

$$\mathcal{L}_{\text{recon}} = \mathbb{E}_{(x_c, a)} \left[ \, \| f_\theta(x_c, a) - x_{\text{target}} \|_2^2 \, \right]. \tag{6}$$

The direction $a$ is encoded as a discrete conditioning token / embedding that modulates generation. Unless otherwise noted, we follow the default loss composition and hyperparameters of the base training recipe, without introducing bespoke auxiliary losses or coefficients. This setup leverages trajectory-based supervision to promote spatial continuity and directional controllability while keeping the objective simple and reproducible.

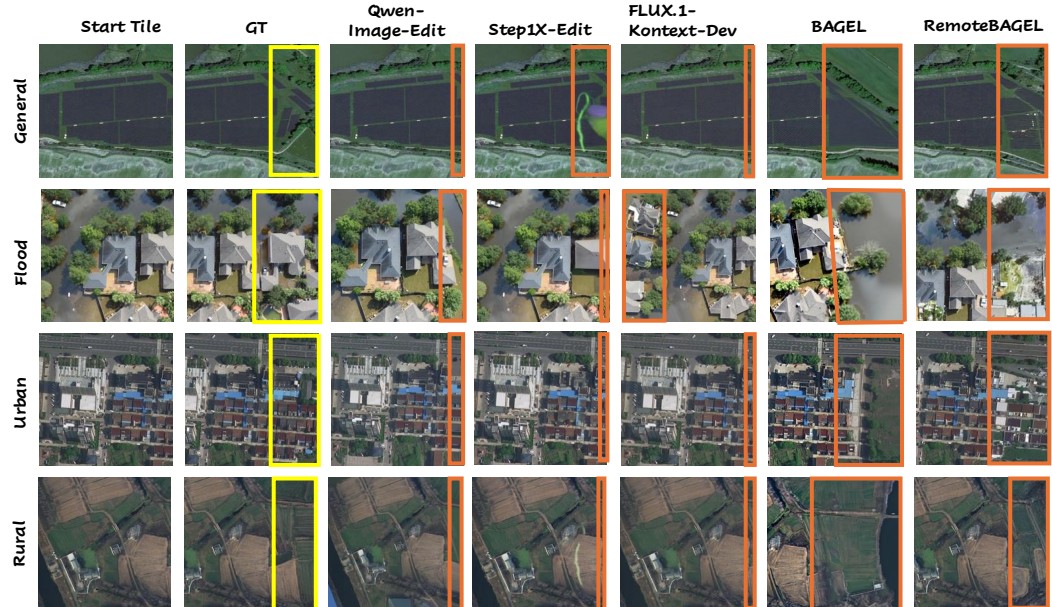

Figure 5: Qualitative comparison of rightward continuations across four scenarios (general, flood, urban, rural). RemoteBAGEL produces geospatially consistent extrapolations aligned with the ground truth, whereas other models often generate invalid or semantically inconsistent content.

## 5 EXPERIMENT AND RESULTS

### 5.1 EXPERIMENTAL SETTING

**Dataset Construction.** All training and evaluation data are strictly separated. Although the dataset sources have been described earlier, here we emphasize that the training set and the RSWISE benchmark share no overlapping images or geographic regions. We manually screened all samples to ensure that no training tile originates from the same or even a visually similar location as any test tile, fully preventing data leakage. To construct the training instances, each image is divided into a $3 \times 3$ grid, producing nine tiles. Any tile that has at least one adjacent tile is treated as a potential starting tile, and each neighboring tile defines a direction-specific transition. Enumerating all valid start–neighbor combinations yields 24 distinct directional actions per image. Across the 420 selected training images, this procedure results in a total of 10,080 action-conditioned training pairs. And the benchmark utilizes 400 unique satellite images, each divided into a $3 \times 3$ grid where the central tile is the starting block. This results in 1,600 test instances, corresponding to the four cardinal direction extrapolation tasks relative to the central starting tile.

**Implementation Details.** We fine-tune `BAGEL-7B` on the 10,080 action-conditioned instances using $4\times$ H100 (80GB) GPUs for approximately 20 hours. For fair comparison, inference for all five models across the full 1,600 benchmark tasks is performed in a strict zero-shot mode, totaling roughly 8,000 runs and requiring about 8 hours of compute using $10\times$ A100 (80GB) GPUs.

### 5.2 MAIN RESULTS

Our evaluation reveals a clear performance hierarchy among the tested models. RemoteBAGEL substantially outperforms all baselines across four benchmark scenarios, achieving near-optimal scores ($\sim$95) in *general* and *rural* settings. This represents a significant advancement over BAGEL (58-64 points), despite BAGEL's strong multimodal foundations. The performance gap demonstrates that domain-specific adaptation is crucial for remote sensing tasks, as generic vision-language models struggle to capture the spatial coherence and structural patterns inherent in satellite imagery. Furthermore, an Out-of-Distribution (OOD) test demonstrates RemoteBAGEL's robust generalization capability in unseen hurricane scenarios, with detailed results provided in Appendix G.

| Model | RSWISE-General | RSWISE-Flood | RSWISE-Urban | RSWISE-Rural | Average |
|---|---|---|---|---|---|
| Qwen-Image-Edit (Wu et al., 2025) | 46.9 | 52.1 | 56.5 | 57.2 | 53.2 |
| FLUX.1-Kontext-Dev (Labs et al., 2025) | 40.0 | 18.7 | 43.7 | 41.8 | 36.1 |
| Step1X-Edit (Liu et al., 2025b) | 51.7 | 17.3 | 58.5 | 55.0 | 45.6 |
| BAGEL (Deng et al., 2025) | 64.3 | 64.2 | 62.3 | 58.7 | 62.4 |
| RemoteBAGEL | **95.7** | **78.0** | **87.3** | **94.3** | **88.8** |

Table 2: Performance of different models on RSWISE. Results are reported across four scenarios (General, Flood, Urban, Rural) and their average.

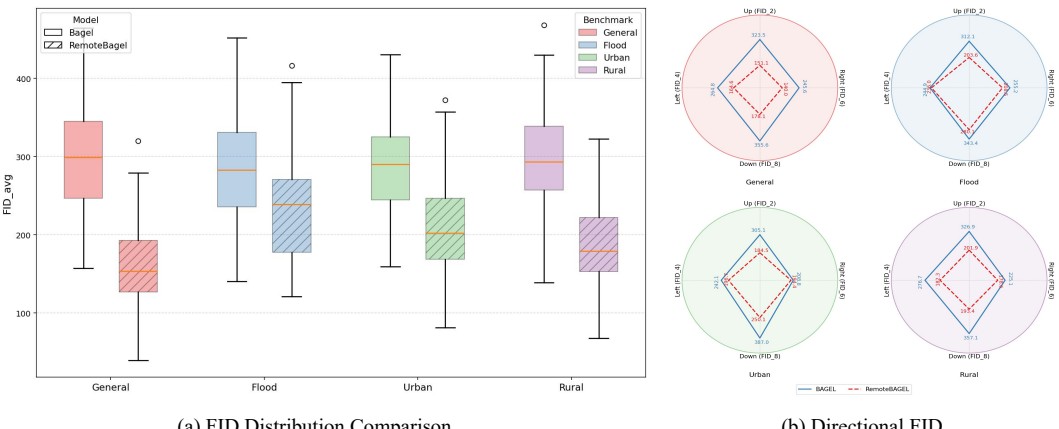

(a) FID Distribution Comparison        (b) Directional FID

Figure 6: FID-based analysis of BAGEL and RemoteBAGEL. (a) Distributional results across four scenarios show consistently lower FID for RemoteBAGEL, with the largest gains in general and rural settings. (b) Directional results reveal an anisotropic pattern, where left/right continuations are easier than up/down, indicating directional bias in spatial extrapolation.

## 5.3 RESULT ANALYSIS

**Failure Mode Analysis**    Qualitative analysis in Figure 5 reveals distinct failure modes across model categories. While BAGEL generates visually plausible outputs, it occasionally violates spatial consistency-producing incorrect orientations or ignoring geometric regularities. However, compared to baselines, its potential for reasoning is evident. Crucially, although baseline approaches also utilize unified architectures, they fail to effectively propagate the commonsense knowledge from their frozen large language model (LLM) backbones into the visual generation process. Consequently, they operate primarily as appearance editors, generating outputs that mimic input textures but lack genuine spatial extrapolation. BAGEL partially overcomes this by leveraging better multimodal alignment, yet it still falters due to a lack of domain-specific grounding; it struggles to map abstract direction tokens to the complex spatial semantics of satellite imagery. In contrast, the superior performance of RemoteBAGEL demonstrates that our fine-tuning strategy successfully bridges this gap. It allows the model to internalize geospatial priors and effectively generalize the backbone's inherent reasoning capabilities to the task of direction-aware spatial world modeling.

**Visual Fidelity Analysis**    Figure 6 (a) compares the FID distributions of BAGEL and Remote-BAGEL across four scenarios. RemoteBAGEL consistently achieves superior visual fidelity with lower FID scores and reduced variance in all settings, though the degree of improvement varies significantly by scenario type. The gains are most pronounced in *general* and *rural* scenarios (FID reductions $> 20\%$), where repetitive agricultural patterns and homogeneous textures benefit substantially from domain-specific training. *Urban* scenarios show moderate but consistent improvements-while geometric regularities in roads and buildings provide structural cues, the higher variance suggests ongoing challenges in capturing fine-grained urban diversity. *Flood* scenarios prove most challenging for both models, exhibiting the smallest improvements due to irregular and dynamic water boundaries that resist systematic pattern learning. These results demonstrate that structured, pattern-rich environments are more suitable to generative modeling than highly variable or transient phenomena.

**Directional Continuation Analysis** Figure 6 (b) illustrates an anisotropic performance pattern where horizontal continuations consistently outperform vertical ones. We attribute this asymmetry to the interplay between physical imaging factors and the model's inductive biases. Physically, variations in solar azimuth and the polar orbital paths of satellites introduce greater radiometric and geometric inconsistencies along the vertical axis, creating naturally harder extrapolation targets compared to the more stable illumination in horizontal directions. From a modeling perspective, this anisotropy likely reflects the distribution of spatial priors in the pre-trained LLM backbone. In general multimodal corpora, horizontal relationships (*e.g.*, reading order, panoramic layouts) are often represented with stronger continuity and concrete logical links than vertical ones. Consequently, the direction tokens for *left/right* may activate more robust spatial reasoning patterns within the model than *up/down* tokens. This combination of inherent domain challenges (lighting/orbit) and model-level inductive bias (weaker vertical priors) results in the observed performance gap.

## 5.4 ABLATION STUDIES

We validate key design choices via controlled ablations (details in Appendix F). Results in Table 3 highlight three findings:

**Prompt Formulation.** Grid-aligned instructions ("up/down") significantly outperform cardinal directions ("north/south"). The model exhibits poor responsiveness to geographic terms, failing to accurately execute the specific directional commands compared to image-relative coordinates.

**Spatial Overlap.** High spatial overlap (66.7%) is fundamental for maintaining geospatial coherence. Reducing overlap disrupts boundary continuity during inference and prevents the model from learning valid cross-boundary transitions during training.

**Directional Conditioning.** Explicit direction tokens are essential. Removing them leads to a drastic performance drop to 58.7, resulting in random generation where the model fails to explicitly generate content according to the instructions.

| Method | RSWISE↑ | FID↓ | GPT↑ |
|---|---|---|---|
| *Inference Phase* | | | |
| *Prompting Strategy* | | | |
| No prompt | 52.3 | 215.4 | 4.10 |
| Cardinal (N/S/E/W) | 72.1 | 201.5 | 7.20 |
| **Grid-aligned (Ours)** | **88.8** | **196.0** | **8.86** |
| *Overlap Ratio* | | | |
| 0% | 78.5 | 235.8 | 8.10 |
| 33% | 82.1 | 215.5 | 8.45 |
| **66.7% (Ours)** | **88.8** | **196.0** | **8.86** |
| *Training Phase* | | | |
| *Directional Conditioning* | | | |
| No direction token | 58.7 | 198.5 | 5.10 |
| **With token (Ours)** | **88.8** | **196.0** | **8.86** |
| *Overlap Strategy* | | | |
| 0% | 74.6 | 235.1 | 6.80 |
| 33% | 80.2 | 210.4 | 7.95 |
| **66.7% (Ours)** | **88.8** | **196.0** | **8.86** |

Table 3: Ablation studies on Inference and Training configurations.

## 5.5 VALIDITY OF GPT-BASED EVALUATION

We confirmed the scientific rigor of our GPT-4o metric via two studies: multi-run stability (mean standard deviation 0.026) and human agreement (Spearman $\rho = 0.72$). These results establish the GPT-based metric as a reliable and expert-aligned semantic evaluator, essential for distinguishing genuine spatial reasoning from visual plagiarism. Detailed statistics are provided in Appendix E.

## 6 CONCLUSION

This work introduces the first framework for world modeling in remote sensing. We formulate direction-conditioned spatial extrapolation as a novel task, establish the RSWISE benchmark with dual-dimension evaluation, and develop RemoteBAGEL, a model that achieves state-of-the-art performance in spatial reasoning. Our findings highlight the potential of world models to capture large-scale geospatial structures in remote sensing with both semantic and distributional consistency. We discuss the future applications and potential impacts in Appendix H.

## ETHICS STATEMENT

This work uses publicly available remote sensing datasets strictly for research purposes. No personally identifiable information (PII) was collected or annotated. All datasets are used in compliance with their licenses. We acknowledge that remote sensing models can potentially be misused (e.g., surveillance or monitoring of sensitive areas). To mitigate these risks, we release code and models for research and educational purposes only, and we discuss potential limitations and responsible use. No human subjects or IRB-regulated studies were involved in this work.

## REPRODUCIBILITY STATEMENT

We will release our dataset, training code, inference code, and trained model checkpoints after acceptance to ensure reproducibility. We also provide detailed descriptions of preprocessing steps, hyperparameter settings, and training schedules in the main text and appendix. The scoring procedure for automatic evaluation is fully documented to allow independent verification.

## LLM USAGE

We did not use large language models (LLMs) for writing, editing, ideation, or literature retrieval. LLMs were only used as evaluation tools in our benchmark experiments. Specifically, we employed GPT as an automatic evaluator, with fixed prompts and documented version information, to ensure reproducibility. The detailed scoring procedure is fully described in the paper.

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

APPENDIX

# A SYSTEM PROMPT FOR REMOTE SENSING WORLD GENERATION EVALUATION

**System Prompt for Remote Sensing Spatial reasoning Evaluation**

You are a strict remote sensing image evaluation specialist. You must evaluate if the generated image (Image B) *faithfully executes the directional instruction* relative to the original image (Image A).

You will be given:
1. Image A: the original remote sensing image.
2. Image B: a generated image based on the instruction.
3. Instruction: a spatial directive (e.g., "look up", "look left", "look right", "look down").

## Evaluation Rules:
- Image B must clearly reveal **substantially new content** in the instructed direction.
- **Small shifts, pixel stretching, zoom-in/out, color tweaks, or tiny border changes DO NOT count as valid compliance.**
- If Image B looks nearly identical to Image A (with only minor edge changes), this must be scored **very low (0–2)**.
- The judgment should prioritize **meaningful new scene exposure** over visual quality.
- High scores are reserved only for cases where Image B shows **distinct, new regions** in the correct direction, beyond what was visible in Image A.

## Reasoning steps:
1. Detect Difference: What substantial new areas or structures appear in Image B compared to Image A?
2. Expected Expansion: Describe what should be revealed if the instruction was correctly followed.
3. Instruction Match: Check if the differences in (1) align with (2).
   - Are the changes large enough to be meaningful?
   - Are the new regions clearly visible, not just stretched borders?
   - Are they in the correct direction?
4. Decision: Assign a score (0–10).

## Scoring scale (0–10):
- **10 (Perfect)**: Significant new regions appear exactly in the correct direction.
- **8–9 (Strong)**: Mostly correct, with slight incompleteness.
- **6–7 (Moderate)**: Some valid new content, but limited or partly misplaced.
- **4–5 (Weak)**: Minimal or ambiguous new content, partial compliance at best.
- **2–3 (Very Weak)**: Only border pixels stretched or tiny changes; not meaningful expansion.
- **0–1 (Non-Compliance)**: No new content, or expansion in the wrong direction.

Output strictly in JSON format:
{
  "instruction_score": X,
  "reasoning": "..."
}

Figure 7: System Prompt for Remote Sensing Spatial Reasoning Evaluation

# B PROMPTS USED IN RSWISE EVALUATION.

For reproducibility, we list the exact textual prompts used in the benchmark (defined in the image-grid frame with up, down, left, and right, rather than geographic cardinal directions).

- "Look up at this picture"
- "Look down at this picture"
- "Look left at this picture"
- "Look right at this picture"

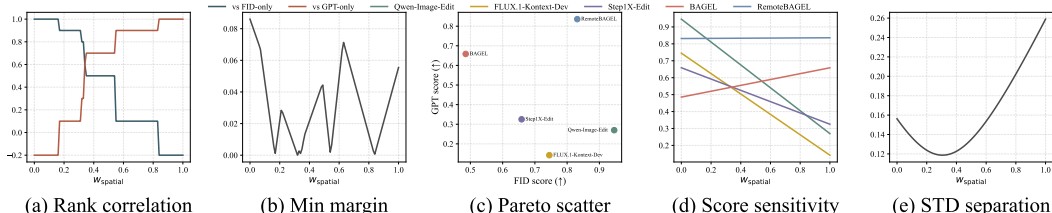

|  (a) Rank correlation | (b) Min margin | (c) Pareto scatter | (d) Score sensitivity | (e) STD separation |

Figure 8: Weight analysis of RSWISE across different criteria. Rankings are stable within $[0.5, 0.7]$, supporting the choice of $w_{\text{spatial}} = 0.6$, $w_{\text{fid}} = 0.4$.

# C    DETAILED WEIGHT ANALYSIS IN RSWISE

**Weight analysis.**    We validate the RSWISE weighting scheme through a systematic scan of $w_{\text{spatial}}$ (Figure 8). The five diagnostics respectively examine rank correlation, min margin, pareto scatter, score sensitivity, and STD separation. Together they show that rankings remain stable and discriminative power is preserved within the interval $[0.5, 0.7]$, supporting the choice of $w_{\text{spatial}} = 0.6$ and $w_{\text{fid}} = 0.4$. Here we provide detailed interpretations of the individual plots.

**Rank correlation.**    The first plot reports the Spearman rank correlation between rankings obtained at each $w_{\text{spatial}}$ and those at the two extreme endpoints (FID-only and spatial-only). As $w_{\text{spatial}}$ increases, rankings gradually shift from being FID-driven to spatial-driven, and stabilize near 0.6, reflecting a balanced compromise.

**Min margin.**    The second plot shows the minimum pairwise gap between adjacent models after sorting by their combined scores. Larger margins indicate stronger discriminative power. Although the margin fluctuates, relatively higher values occur around $w_{\text{spatial}} = 0.6$, suggesting reliable separation in this region.

**Pareto scatter.**    The third plot compares models in terms of their mean FID scores (realism) and mean spatial scores (semantic continuation). Different models occupy distinct positions on the Pareto front—for instance, Qwen-Image-Edit aligns more with realism, whereas BAGEL emphasizes semantic continuation. This demonstrates the complementarity of the two metrics and supports the need for a mixed weighting scheme.

**Score sensitivity.**    The fourth plot depicts how overall scores for each model vary with $w_{\text{spatial}}$. Although absolute values change, the relative ordering of models remains stable across $[0.5, 0.7]$, indicating that weights within this interval do not affect qualitative comparisons.

**STD separation.**    The fifth plot presents the standard deviation of scores across models. While separation grows steadily toward spatial-only weighting, discarding FID entirely would eliminate grounding in reference realism. We therefore restrict the range to $[0.4, 0.8]$ and apply a mild prior near 0.6.

**Conclusion.**    Collectively, these diagnostics show that $[0.5, 0.7]$ is a robust interval where rankings remain stable and margins acceptable. The unconstrained optimum lies close to $w_{\text{spatial}} = 0.63$, but for clarity and reproducibility we finalize $w_{\text{spatial}} = 0.6$ and $w_{\text{fid}} = 0.4$, consistent with both data-driven analysis and interpretability considerations.

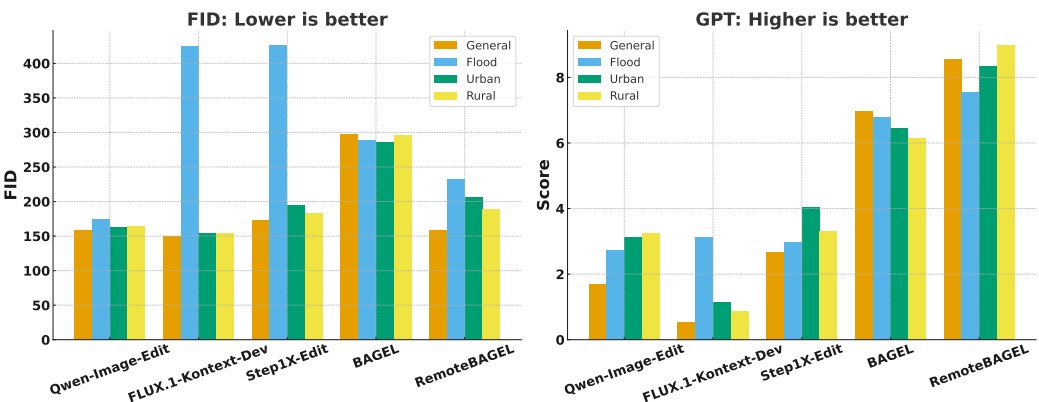

Figure 9: Comparison of models on raw FID (lower is better) and GPT (higher is better).

# D  FULL METRIC RESULTS

Table 4 presents the complete results of our evaluation metrics across four scenarios (general, flood, urban, rural). RSWISE serves as the overall aggregated score, while FID and GPT correspond to its constituent sub-scores. The arrows indicate performance trends, with ↓ meaning lower is better and ↑ meaning higher is better.

| Metric / Scenario | Qwen-Image-Edit | FLUX.1-Kontext-Dev | Step1X-Edit | BAGEL | RemoteBAGEL |
|---|---|---|---|---|---|
| **RSWISE (↑)** | | | | | |
| General | 46.9 | 40.0 | 51.7 | 64.3 | **95.7** |
| Flood | 52.1 | 18.7 | 17.3 | 64.2 | **78.0** |
| Urban | 56.5 | 43.7 | 58.5 | 62.3 | **87.3** |
| Rural | 57.2 | 41.8 | 55.0 | 58.7 | **94.3** |
| Average | 53.2 | 36.1 | 45.6 | 62.4 | **88.8** |
| **FID (↓)** | | | | | |
| General | 157.96 | 149.90 | 173.14 | 297.39 | **158.44** |
| Flood | 173.84 | 424.60 | 426.29 | 288.88 | **232.06** |
| Urban | 163.40 | 153.43 | 194.63 | 285.77 | **206.42** |
| Rural | 164.10 | 153.75 | 182.75 | 296.46 | **189.06** |
| **GPT (↑)** | | | | | |
| General | 1.6775 | 0.5388 | 2.6575 | 6.9673 | **8.5489** |
| Flood | 2.7300 | 3.1404 | 2.9750 | 6.7750 | **7.5600** |
| Urban | 3.1400 | 1.1375 | 4.0550 | 6.4400 | **8.3475** |
| Rural | 3.2400 | 0.8725 | 3.3183 | 6.1575 | **8.9825** |

Table 4: Comparison of models across RSWISE, FID, and GPT benchmarks after transposing (rows = metrics/scenarios, columns = models). Arrows indicate direction of better performance (↑ higher is better, ↓ lower is better).

| Metric (Avg.) | Qwen-Image-Edit | FLUX.1-Kontext-Dev | Step1X-Edit | BAGEL | RemoteBAGEL |
|---|---|---|---|---|---|
| RSWISE (↑) | 53.2 | 36.1 | 45.6 | 62.4 | **88.8** |
| FID (↓) | 164.8 | 254.7 | 244.7 | 292.1 | **196.0** |
| GPT (↑) | 2.70 | 1.42 | 3.25 | 6.59 | **8.86** |

Table 5: Average performance of different models across three benchmarks: RSWISE, FID, and GPT. Arrows indicate direction of better performance (↑ higher is better, ↓ lower is better).

## D.1  FID VS. GPT METRICS.

Figure 9 highlights the complementary roles of the two metrics. Raw FID, dominated by texture and artifact statistics, offers a reliable measure of visual fidelity and detail preservation, but is less

sensitive to semantic plausibility or directional accuracy. GPT-based evaluation, in contrast, directly probes spatial reasoning by scoring continuity, transitions, and compliance with the instructed direction, yet is less attuned to subtle degradations in low-level image quality. This explains why generic editing baselines such as Qwen-Image-Edit and FLUX.1-Kontext-Dev achieve competitive FID values despite failing to introduce meaningful extrapolated content, as reflected in their low GPT scores. At the same time, RemoteBAGEL attains the highest GPT scores across all scenarios (up to 8.98 in *rural*) while maintaining competitive FID even in challenging settings such as *flood*. Together, GPT captures the decisive dimension of semantic extrapolation, while FID provides complementary sensitivity to visual quality, sharpening distinctions between strong models like BAGEL and RemoteBAGEL.

# E   VALIDITY AND RELIABILITY OF GPT-BASED EVALUATION

To ensure robustness, we conducted two additional studies: a multi-run reproducibility analysis and a large-scale human–GPT agreement study. Both results confirm that our evaluation protocol is stable and aligned with expert judgment.

## E.1   REPRODUCIBILITY: MULTI-RUN STABILITY ANALYSIS

To quantify the impact of potential randomness in GPT-4o scoring, we re-evaluated the full RSWISE benchmark using **five independent GPT-4o runs**. The results, reported in Table 6, demonstrate **very high stability**:

- The mean standard deviation across all tasks is **0.026** on the 0–10 scale.
- **88.2%** of samples receive *identical* scores across all runs.
- Only **1.1%** of samples exhibit a variance of 2 points or more.

Crucially, *no model ranking changed* when averaging across runs. This confirms that GPT-4o-based scores are sufficiently reproducible for scientific benchmarking.

| Scope | Mean Std Dev | Exact Match | Minor Variation | Major Variation |
|---|---|---|---|---|
| All Tasks | 0.026 | 88.2% | 10.7% | 1.1% |

Table 6: Multi-run stability of GPT-4o spatial reasoning scores (5 runs). We report the mean standard deviation and the percentage of samples exhibiting exact matches or score variations.

## E.2   VALIDITY: AGREEMENT BETWEEN HUMAN EXPERTS AND GPT-4O

To verify semantic correctness, we conducted a **2,000-sample human evaluation study**, balanced across all models, scenarios, and directions. Five remote-sensing experts scored each sample using the same rubric as GPT-4o.

The results (Table 7) show a strong alignment (Spearman $\rho = 0.72$) with an overall Mean Absolute Error (MAE) of **0.86**. Notably:

- GPT-4o scores of **9–10** correspond to a human expert mean of **9.1**.
- GPT-4o scores of **0** correspond to a human expert mean of **0.9**.

This demonstrates strong alignment, especially at the high-quality and failure extremes, confirming that GPT-4o reliably captures spatial semantic correctness.

# F   ABLATION STUDIES

We address concerns regarding design choices using controlled ablations. These experiments isolate the impact of prompting and spatial overlap during **inference** and **training**.

| GPT Score Range | Proportion | Human Mean | Human Std. | MAE |
|---|---|---|---|---|
| 0 | 12% | 0.9 | 1.1 | 0.9 |
| 1–2 | 15% | 2.3 | 1.4 | 1.1 |
| 3–4 | 20% | 4.2 | 1.6 | 1.2 |
| 5–6 | 21% | 5.8 | 1.4 | 1.1 |
| 7–8 | 17% | 7.4 | 1.2 | 0.8 |
| 9–10 | 15% | 9.1 | 0.9 | 0.9 |
| **Overall** | 100% | — | — | 0.86 |
| Spearman $\rho$ (GPT vs. human mean) | | | | 0.72 |

Table 7: Agreement between human experts and GPT-4o. We report the proportion of samples in each GPT score bucket, along with human statistics and MAE.

## F.1 INFERENCE-TIME ABLATIONS

**Prompt Formulation.** We evaluate three prompting strategies using the same trained Remote-BAGEL model: (1) **No direction prompt**, (2) **Cardinal prompts** ("north", "south", etc.), and (3) **Grid-aligned prompts** ("up", "down", etc.; *ours*). As shown in Table 8, grid-aligned prompts provide the strongest conditioning. Removing the prompt collapses spatial reasoning, while cardinal prompts are less effective.

| Prompt Formulation | RSWISE ↑ | FID ↓ | GPT Score ↑ |
|---|---|---|---|
| No prompt | 52.3 | 215.4 | 4.10 |
| Cardinal (N/S/E/W) | 72.1 | 201.5 | 7.20 |
| **Grid-aligned (Ours)** | **88.8** | **196.0** | **8.86** |

Table 8: Inference ablation: Effect of Prompt Formulation.

**Evaluation Overlap.** We test 0%, 33%, and 66.7% spatial overlap during inference. Larger overlap consistently improves spatial continuity (Table 9). High-overlap crops provide the necessary context for resolving boundary consistency without diminishing extrapolation difficulty, making 66.7% the optimal choice.

| Overlap Ratio | RSWISE ↑ | FID ↓ | GPT Score ↑ |
|---|---|---|---|
| 0% | 78.5 | 235.8 | 8.10 |
| 33% | 82.1 | 215.5 | 8.45 |
| **66.7% (Ours)** | **88.8** | **196.0** | **8.86** |

Table 9: Inference ablation: Effect of Overlap Ratio.

## F.2 TRAINING-TIME ABLATIONS

**Directional Conditioning.** We retrain RemoteBAGEL **with** and **without** direction tokens. Table 10 confirms that without directional conditioning, the model cannot distinguish between different extrapolation directions, leading to generic outpainting behaviors.

| Training Setup | RSWISE ↑ | FID ↓ | GPT Score ↑ |
|---|---|---|---|
| No direction token | 58.7 | 198.5 | 5.10 |
| **With direction token (Ours)** | **88.8** | **196.0** | **8.86** |

Table 10: Training ablation: Influence of Directional Conditioning.

**Training Overlap.** We ablate the overlap ratio used for constructing training crops (Table 11). Sufficient overlap during fine-tuning is necessary for the model to learn cross-boundary transition patterns, validating the use of 66.7% overlap in our design.

| Training Overlap | RSWISE ↑ | FID ↓ | GPT Score ↑ |
|---|---|---|---|
| 0% | 74.6 | 235.1 | 6.80 |
| 33% | 80.2 | 210.4 | 7.95 |
| **66.7% (Ours)** | **88.8** | **196.0** | **8.86** |

Table 11: Training ablation: Effect of Overlap Ratio.

**Summary.** Across all ablations, the conclusions are consistent: **Grid-aligned directions** are empirically optimal for conditioned extrapolation, and **high spatial overlap** is required both during training and inference to ensure coherent boundary transitions. These components are not incidental design choices—they are necessary ingredients for reliable spatial world modeling.



Figure 10: Qualitative example of RemoteBAGEL spatial extrapolation on an unseen hurricane-impacted urban region. The model successfully continues geospatial structures despite severe appearance shifts (damage).

## G OUT-OF-DISTRIBUTION ROBUSTNESS EVALUATION

To further validate the generalization capabilities of RemoteBAGEL, we performed an **Out-of-Distribution (OOD)** experiment using satellite imagery from a previously unseen disaster scenario: a hurricane-impacted urban region. This dataset is completely disjoint from the training distribution (Sky-SA, FloodNet, LoveDA) and the RSWISE benchmark imagery. The experiment consisted of evaluating RemoteBAGEL in a fully zero-shot manner on 100 randomly sampled OOD tasks, following the established RSWISE protocol.

| Metric | OOD Mean Score |
|---|---|
| **RSWISE** | **83.7** |
| **FID** | **208.6** |
| **GPT Score** | **8.41** |

Table 12: RemoteBAGEL OOD performance on unseen Hurricane-Disaster imagery (100 tasks).

**Analysis and Interpretation.** The results, summarized in Table 12, demonstrate that RemoteBAGEL maintains strong and reliable spatial reasoning even under severe appearance shifts characteristic of hurricane damage. The aggregated RSWISE score of **83.7** and the high GPT Spatial Score of 8.41 show only modest degradation relative to the in-domain average ($\sim 88.8$).

Qualitative inspection (see Figure 10) further confirms this resilience, showing correct continuation of primary geospatial structures, including damaged road networks, complex coastline morphology, and clusters of damaged buildings. This performance confirms that RemoteBAGEL is not merely memorizing tile patterns or dataset-specific textures; rather, it has learned generalizable geospatial continuity priors that effectively transfer to an entirely new, challenging disaster scenario. This result strengthens our core claim that direction-conditioned spatial extrapolation captures transferable world-model–style structure, rather than dataset-specific tile completion.

## H FUTURE APPLICATIONS AND POTENTIAL IMPACTS

Our work introduces **RemoteBAGEL** and the **RSWISE benchmark** to establish the foundation for **world modeling in remote sensing** through **direction-conditioned spatial extrapolation**. While our task does not replace physically-grounded simulations or established planning tools, its capability to capture **geospatial continuity** and **latent structural regularities** across large-scale satellite imagery opens several avenues for future research and integration into complex downstream systems.

### H.1 FUTURE DIRECTIONS: TOWARDS COMPREHENSIVE GEOSPATIAL MODELING

The subsequent direction is to leverage our foundational spatial capability to build toward more comprehensive geospatial world models, focusing on integrating the missing complexity.

- **Complementary Geospatial Prior:** The learned latent representation of spatial continuity serves as a **structural cue** or **weak prior**, and can be fused with time-series data, physical variables (e.g., DEM, rainfall), and SAR backscatter. This integration aims to provide a robust **spatial viewpoint** that complements the temporal dynamics or physical constraints of operational models.

- **Data Augmentation and Gap Filling:** The model's ability to generate plausible spatial continuations can be used for intelligent data imputation, helping to fill gaps or occlusions in expansive satellite imagery datasets where direct observation is unavailable.

- **Advanced Evaluation Metrics:** Future work will build upon RSWISE by developing more advanced evaluators, such as consistency checks, structural topology metrics, and attention-based diagnostics, to provide a more holistic definition of spatial reasoning.

### H.2 SUPPORTING DOWNSTREAM APPLICATIONS

While not performing end-to-end prediction, the learned spatial priors can enhance specific stages of high-impact applications:

- **Enhanced Flood & Disaster Modeling:** In flood-related applications, integrating the inferred plausible continuation of flooded or at-risk regions (as captured by the model) can offer an **auxiliary spatial cue** on propagation patterns. This assists in providing a richer input for operational **hydrological models** that depend on parameters like flood depth and physical simulations, rather than replacing them.

- **Augmenting Urban & Infrastructure Planning:** For urban-planning scenarios, our model captures **latent morphological regularities** (e.g., road network topology, block structure continuity). These structural insights can serve as valuable **auxiliary cues** when combined with traditional GIS layers, socioeconomic indicators, and urban form data, helping to inform growth models or infrastructure monitoring systems.

## I SPATIAL EXTRAPOLATION PERFORMANCE

Among the baselines, several models produce limited or near-trivial continuations, whereas BAGEL generates richer content but with a higher incidence of semantic hallucinations, stylistic drift, and viewpoint misalignment. Within our experimental setup and datasets—and as reflected by RSWISE, FID, GPT, and qualitative inspection—RemoteBAGEL achieves the most favorable balance between generation diversity and spatial/semantic consistency, yielding varied yet structurally coherent continuations.

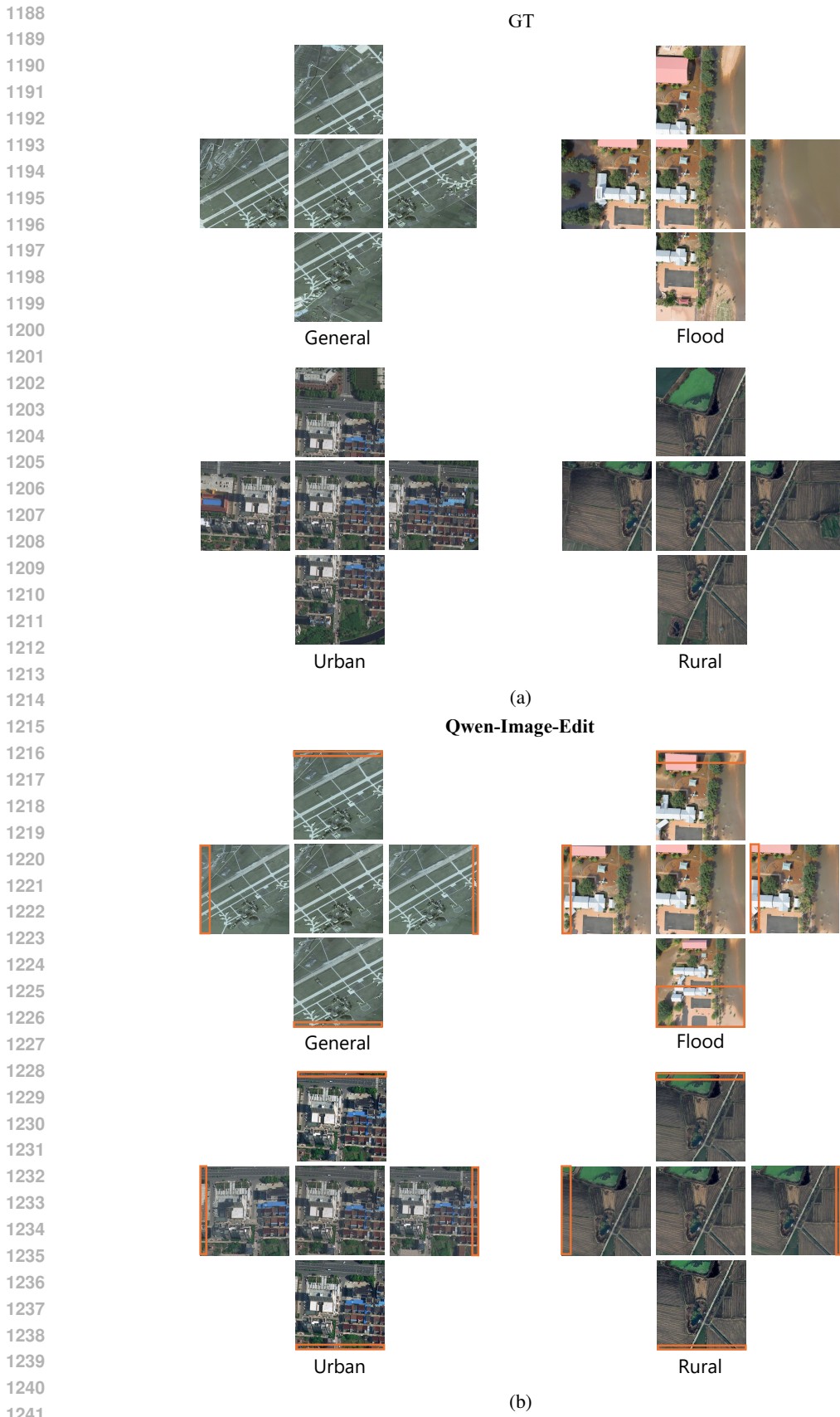

Figure 11: Spatial extrapolation performance of five models across four scenarios and four directions (up, down, left, right).

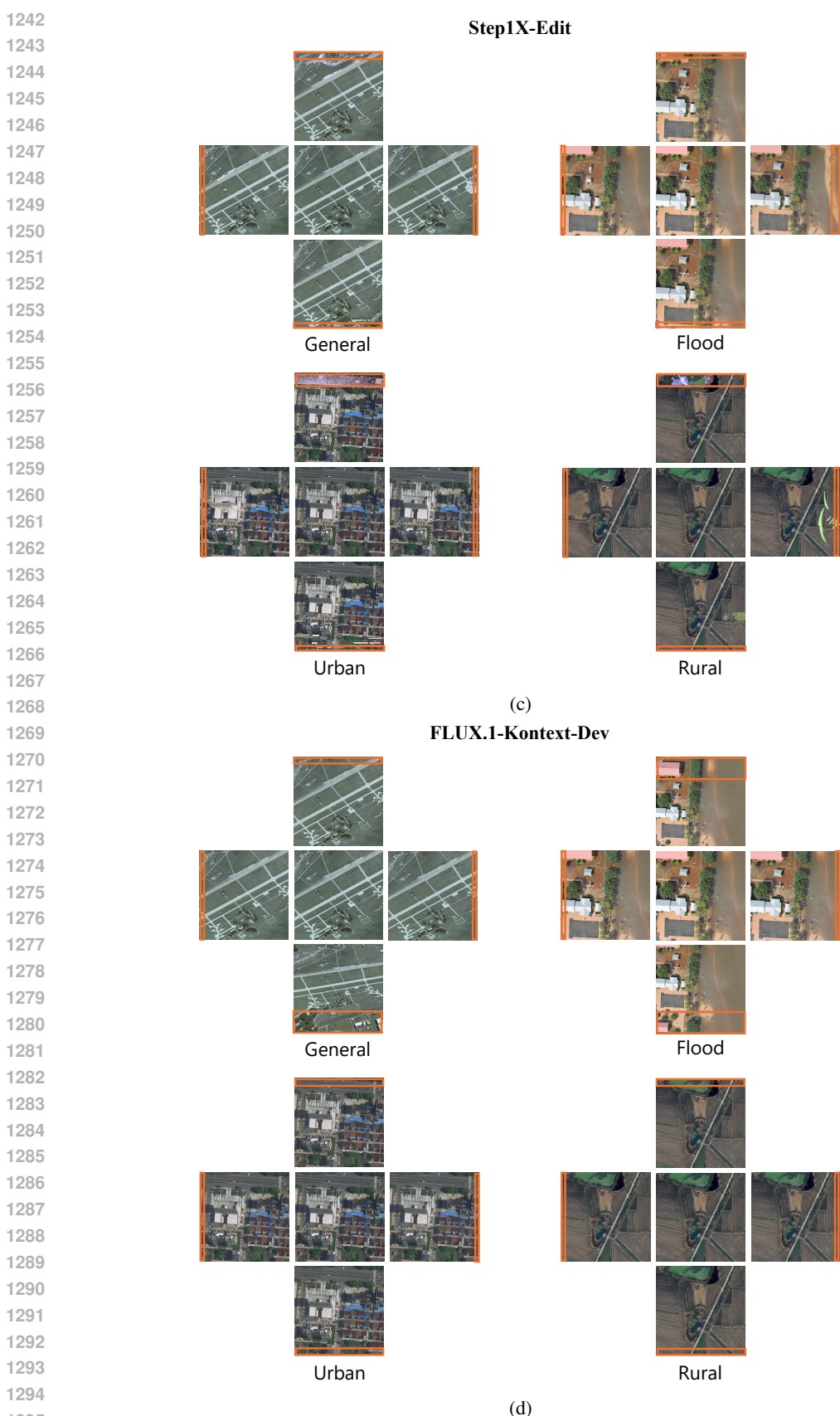

Figure 11: Spatial extrapolation performance of five models across four scenarios and four directions (up, down, left, right).

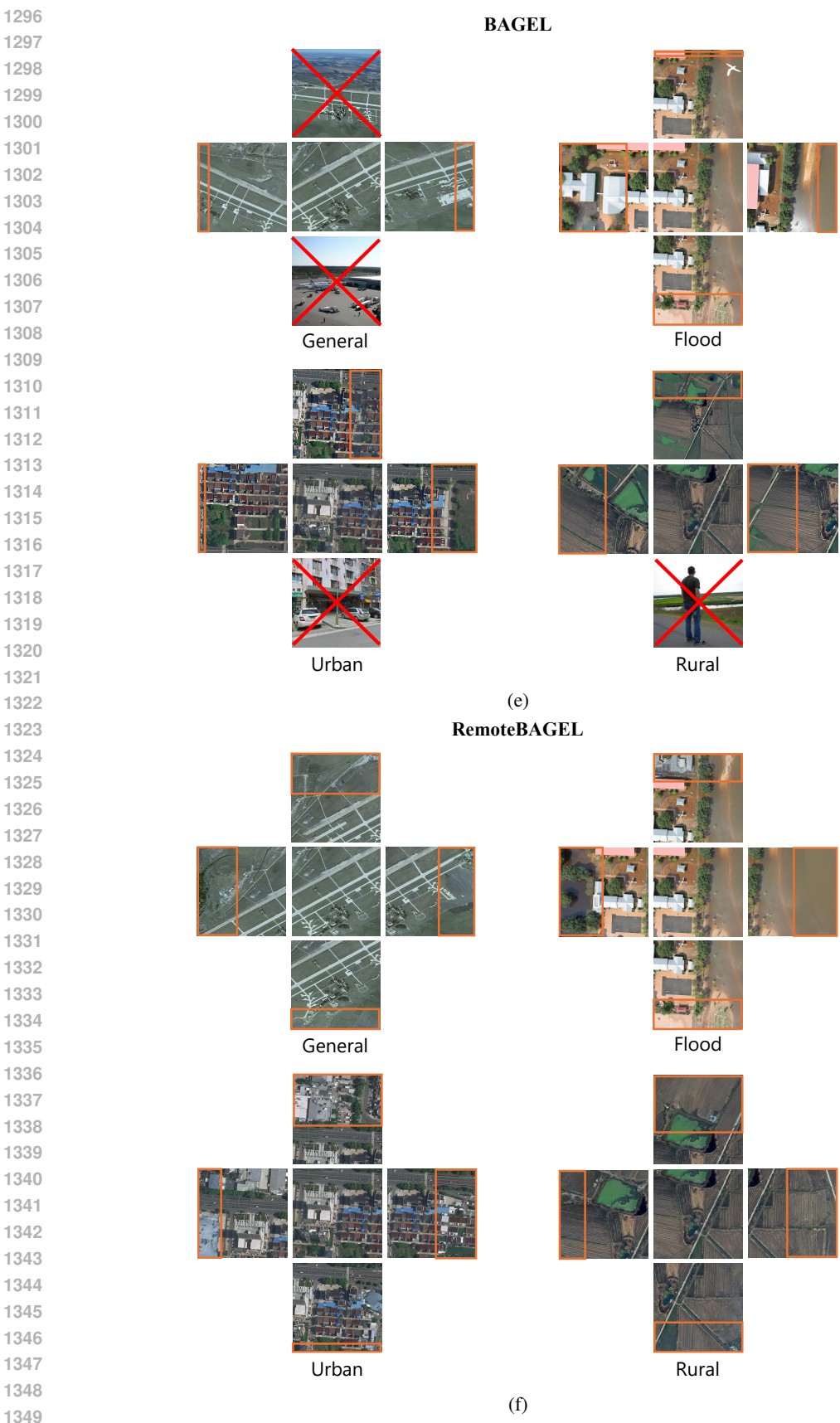

Figure 11: Spatial extrapolation performance of five models across four scenarios and four directions (up, down, left, right).

