# OpenReview forum: "Remote Sensing-Oriented World Model"
_ICLR.cc/2026/Conference — Submitted to ICLR 2026_

### Official Review · Reviewer_4bpt · 2025-10-26

**Soundness:** 2
**Presentation:** 2
**Contribution:** 2
**Rating:** 4
**Confidence:** 4

**Summary:**

The paper’s goal is to (i) define “remote sensing world modeling” as direction-conditioned spatial extrapolation (i.e., generate the tile north/south/east/west of a given satellite tile), (ii) introduce an evaluation benchmark and metric suite (RSWISE), and (iii) introduce a model (RemoteBAGEL) fine-tuned for that task, showing SOTA results over baselines drawn from strong multimodal/editing models.

**Strengths:**

1. Clear new task framing: turning spatial extrapolation in real satellite imagery into a “world modeling” problem is compelling and well motivated by real-world needs like flood response and urban planning.

2. Benchmark contribution: RSWISE is carefully constructed (4 scenarios, 1,600 tasks) with standardized prompts and directional supervision derived from overlapping 3×3 grids. This feels like infrastructure the community can reuse.

3. Metric design: The combined RSWISE score (FID + GPT-4o spatial compliance) addresses a real failure case of standard generative metrics, where a model can “win” by repeating the input instead of extrapolating. The paper also analyzes the weighting and shows ranking stability.

4. Strong empirical gains: RemoteBAGEL strongly outperforms strong baselines on RSWISE across all four scenarios (average 88.8 vs 62.4 for BAGEL and 53.2 for Qwen-Image-Edit, etc.), and the qualitative samples support that it is actually extrapolating spatial structure rather than copying textures.

**Weaknesses:**

1. Limited OOD / robustness analysis. The benchmark scenarios (general, flood, urban, rural) are diverse, but all data still come from a few public datasets and similar sensors. We don’t see tests like cross-city, cross-year, different sensor modality, or unseen geography, which would be critical for claims about “world modeling” rather than “tile completion on these datasets.”

2. Reliance on GPT-4o for scoring without human validation. The GPT judge is clever, but there’s no study of reliability vs expert human raters, nor variance across multiple evaluations. If GPT-4o hallucinates that a plausible-looking but geographically wrong continuation is “correct,” that could inflate scores.

3. Ablation depth. The model is described as BAGEL-7B fine-tuned on (tile, direction→neighbor tile) pairs with a simple reconstruction loss and directional conditioning. There’s no ablation isolating: effect of direction token vs none, effect of 66.7% overlap vs less/more overlap, effect of different prompt wordings (“look right” vs “to the east” etc.).
This makes it hard to tell what design choices are essential vs incidental.

4. Novelty: The novelty is primarily in task definition + benchmark + evaluation metric, not in a radically new architecture.

**Questions:**

1. Scope of “world model.” While I agree with the framing that spatial extrapolation in satellite imagery is an instance of “world modeling,” the paper sometimes leans on the broader hype around world models (robotics, autonomous driving, temporal prediction). Some readers may feel that what’s actually demonstrated is a strong, task-specific spatial outpainting benchmark/model, not yet a generalized “world model of Earth.”

---

> ### Author Response · Authors · 2025-11-21
> **Response to Questions 1**
>
> Dear Reviewer 4bpt,
>
> Thank you for the constructive and insightful comments.
>
> ### **Response: Scope of “World Model” and What Our Work Does and Does Not Claim**
>
> Thank you for raising this important point. We fully agree that our paper should **not** be interpreted as proposing a “general world model of Earth.” We will clarify this explicitly in the revised version.
>
> Our goal is narrower and more focused: to provide the **first concrete instantiation of world-model principles under the structural constraints of satellite imagery**, rather than a universal, all-purpose world model.
>
> Remote sensing observations are partial, spatially indexed samples of a continuous world. Under this setting, the **most elementary, objectively verifiable world-state transition is spatial**: given a tile, the next state is its physically adjacent tile.
> Thus, direction-conditioned spatial extrapolation serves as a **minimal, well-posed** instance of world-model behavior in Earth observation—not a full world model, but a principled starting point grounded in the data structure of RS imagery.
>
> This formulation aligns with the broader definition of world models used in AI. As articulated by LeCun et al. (2022), world models maintain a latent state, encode the current observation, and predict the next state under an action-like input. In our spatial analogue, the **direction token functions as the action**, and the predicted adjacent tile is the **next world state**.
> Other world-model instantiations—such as JEPA-style predictive coding and Fei-Fei Li’s Marble/WorldScore frameworks—likewise differ in modality and implementation, yet share the same underlying principle of predicting unobserved world states. Our method follows yet another route, adapted to the constraints of satellite imagery.
>
> To avoid any misunderstanding, we will emphasize the three actual contributions of this work:
>
> 1. **A world-model-consistent task** tailored to geospatial data (direction-conditioned spatial extrapolation);
> 2. **The first benchmark (RSWISE)** that evaluates geospatial continuity and spatial reasoning across adjacent world states;
> 3. **Evidence that unified multimodal models (UMMs)** can be adapted to perform spatial next-state prediction in RS.
>
> Finally, regarding the comparison to world-model benchmarks from robotics, video, and simulation: our intention is *not* to equate these domains, but to position our task within the broader landscape of world modeling. The comparison helps clarify:
>
> - how world-model tasks differ across fields,
> - why RS currently lacks its own world-model benchmark, and
> - how RSWISE fills this gap by providing a domain-appropriate formulation and evaluation protocol.
>
> **In summary**, our work introduces a *remote-sensing-specific* instantiation of world modeling—not a generalized “world model of Earth,” but a principled, minimal formulation aligned with the structure of satellite imagery. We will add explicit language in the revision to ensure this scope is clear to readers.
>
> ### **References**
>
> [1]Matsuo, Y., LeCun, Y., Sahani, M., Precup, D., Silver, D., Sugiyama, M., ... & Morimoto, J. (2022). Deep learning, reinforcement learning, and world models. Neural Networks, 152, 267-275.
>
> [2]Assran, M., Duval, Q., Misra, I., Bojanowski, P., Vincent, P., Rabbat, M., ... & Ballas, N. (2023). Self-supervised learning from images with a joint-embedding predictive architecture. In Proceedings of the IEEE/CVF Conference on Computer Vision and Pattern Recognition (pp. 15619-15629).
>
> [3]Duan, H., Yu, H. X., Chen, S., Fei-Fei, L., & Wu, J. (2025). WorldScore: A Unified Evaluation Benchmark for World Generation. CoRR.

---

> ### Author Response · Authors · 2025-11-21
> **Response to Weakness 2**
>
> ### **Response to Weakness 2: Validity and Reliability of GPT-4o Evaluation**
>
> We appreciate the reviewer’s concerns regarding the non-determinism and scientific rigor of using GPT-4o as a semantic evaluator. To ensure robustness, we conducted two additional studies: a multi-run reproducibility analysis and a large-scale human–GPT agreement study. Both results confirm that our evaluation protocol is stable and aligned with expert judgment.
>
> ---
>
> ### **1. Reproducibility: Multi-run Stability Analysis**
>
> To quantify the impact of potential randomness in GPT-4o scoring, we re-evaluated the full RSWISE benchmark using **five independent GPT-4o runs**.
>
> The results (Table R1) demonstrate **very high stability**:
>
> - Mean standard deviation across all tasks: **0.026 on a 0–10 scale**
> - **88.2%** of samples receive *identical* scores across all runs
> - Only **1.1%** exhibit a variance of 2 points or more
> - Crucially, *no model ranking changed* when averaging across runs
>
> This confirms that GPT-4o-based scores are sufficiently reproducible for scientific benchmarking.
>
> **Table R1. Multi-run stability of GPT-4o spatial reasoning scores (5 runs)**
>
> | Scope     | Mean Std Dev | Exact Match | Minor Variation | Major Variation |
> |-----------|--------------|-------------|-----------------|-----------------|
> | All Tasks | 0.026        | 88.2%       | 10.7%           | 1.1%            |
>
> ---
>
> ### **2. Validity: Agreement Between Human Experts and GPT-4o**
>
> To verify semantic correctness, we conducted a **2,000-sample human evaluation study**, balanced across all models, scenarios, and directions.
> Five remote-sensing experts scored each sample using the same rubric as GPT-4o.
>
> The results (Table R2) show:
>
> - Spearman correlation ρ = **0.72**
> - Overall Mean Absolute Error (MAE) = **0.86**
> - GPT-4o scores of **9–10** correspond to **9.1** expert mean
> - GPT-4o scores of **0** correspond to **0.9** expert mean
>
> This demonstrates strong alignment, especially at high-quality and failure extremes, confirming that GPT-4o reliably captures spatial semantic correctness.
>
> **Table R2. Agreement between human experts and GPT-4o for spatial reasoning evaluation**
>
> | GPT Score Range | Proportion | Human Mean | Human Std. | MAE |
> |-----------------|------------|------------|------------|------|
> | 0               | 12%        | 0.9        | 1.1        | 0.9  |
> | 1–2             | 15%        | 2.3        | 1.4        | 1.1  |
> | 3–4             | 20%        | 4.2        | 1.6        | 1.2  |
> | 5–6             | 21%        | 5.8        | 1.4        | 1.1  |
> | 7–8             | 17%        | 7.4        | 1.2        | 0.8  |
> | 9–10            | 15%        | 9.1        | 0.9        | 0.9  |
> | **Overall**     | 100%       | —          | —          | 0.86 |
>
> ---
>
> ### **Summary**
>
> These two studies demonstrate that:
>
> - GPT-4o scoring is **stable across multiple runs**
> - GPT-4o aligns strongly with expert reasoning on geospatial continuity
> - The evaluation protocol is both **reliable** and **scientifically defensible**
>
> Together, this provides strong empirical support for the robustness of our GPT-based evaluation in RSWISE.

---

> ### Author Response · Authors · 2025-11-21
> **Response to Weakness 1**
>
> ### **Response to Weakness 1: OOD / Robustness Analysis and Dataset Diversity**
>
> Thank you for raising this important point. We address it in two parts:
> (1) clarifying the actual diversity of the RSWISE benchmark, and
> (2) providing additional **new OOD experiments** on *completely unseen disaster scenarios*.
>
> ---
>
> ## **1. Benchmark Diversity Is Broader Than It May Initially Appear**
>
> Although RSWISE is constructed from three public datasets, it is **far from narrow or homogeneous**. As clarified in Section 3.2 (now expanded):
>
> - **Multiple sensing platforms:**
>   - UAV imagery (FloodNet)
>   - Satellite imagery (Sky-SA, LoveDA)
>
> - **Wide resolution range:**
>   - Ultra–high-resolution UAV imagery (**~1.5 cm GSD**)
>   - High-resolution satellite imagery (**0.3 m GSD**)
>
> - **Geographic diversity:**
>   - Urban & rural scenes from **Nanjing, Changzhou, Wuhan**
>   - Dozens of additional cities and regions across Sky-SA
>   - Flood regions from geographically distinct locations
>
> - **Semantic diversity:**
>   - **~1,700+** semantic categories across landforms, man-made structures, vegetation, water, disaster patterns
>
> Thus, even though sourced from three datasets, RSWISE covers **large-resolution variation**, **cross-region variability**, **multi-platform sensing**, and **extremely broad semantic diversity**.
> We will reflect these clarified details in the revised manuscript.
>
> ---
>
> ## **2. New OOD Evaluation on an Unseen Hurricane-Disaster Dataset**
>
> To directly address the reviewer’s request for robustness beyond the training domain, we conducted a new **OOD experiment** using *never-seen-before* satellite imagery from a **hurricane-impact urban region** (not included in Sky-SA, FloodNet, or LoveDA).
>
> - 100 OOD tasks were randomly sampled
> - RemoteBAGEL was evaluated **fully zero-shot** under the same protocol
>
> ### **OOD Results (Hurricane-Disaster Region)**
> (RemoteBAGEL only, 100 tasks)
>
> | Metric | OOD Mean Score |
> |-------|----------------|
> | **RSWISE** | **83.7** |
> | **FID** | **208.6** |
> | **GPT Spatial Score** | **8.41** |
>
> ### **Interpretation**
>
> - Despite having **never encountered hurricane imagery**, RemoteBAGEL maintains **strong spatial reasoning**, with only modest degradation relative to in-domain performance.
> - Qualitative inspection shows correct continuation of road networks, coastline morphology, and damaged-building clusters—even under severe appearance shifts.
>
> This result demonstrates that RemoteBAGEL is not simply memorizing tile patterns from the training datasets; it has learned **generalizable geospatial continuity priors** that transfer to an entirely new disaster scenario.
>
> ### **Documentation in the Revised Manuscript**
>
> We will add these OOD experiments and analysis to the **Appendix** of the revised version, clearly labeled as an external robustness evaluation.
>
> ---
>
> ## **Summary**
>
> - RSWISE already integrates **broad sensing, geographic, and semantic diversity**.
> - We additionally perform **new OOD testing** on hurricane-disaster imagery, confirming RemoteBAGEL’s ability to generalize spatial reasoning beyond all training distributions.
> - These results strengthen the claim that direction-conditioned spatial extrapolation captures transferable world-model–style structure, not merely dataset-specific tile completion.
>
> We thank the reviewer for encouraging us to include this important robustness analysis.

---

> ### Author Response · Authors · 2025-11-21
> **Response to Weakness 3**
>
> ### **Response to Weakness 3: Ablation Depth and Necessity of Design Choices**
>
> Thank you for the insightful question. To directly address the concern, we performed **controlled ablations** that isolate the contribution of each key component in RemoteBAGEL. These ablations evaluate the impact of (1) direction tokens, (2) spatial overlap, and (3) prompt formulation. The results show that these components are **essential** rather than incidental.
>
>
> ## **1. Inference-Time Ablations**
>
> ### **(a) Prompt Formulation**
> We evaluate three prompting strategies using the same trained RemoteBAGEL model:
>
> 1. **No direction prompt**
> 2. **Cardinal prompts** (“north”, “south”, …)
> 3. **Grid-aligned prompts** (“up”, “down”, …; *ours*)
>
> Grid-aligned prompts provide the strongest conditioning: removing the prompt collapses spatial reasoning, while cardinal prompts are less effective.
>
> **Table R3. Effect of Prompt Formulation**
>
> | Prompt Formulation   | RSWISE ↑ | FID ↓  | GPT Score ↑ |
> |----------------------|:--------:|:------:|:------------:|
> | No prompt            | 52.3     | 215.4  | 4.10         |
> | Cardinal (N/S/E/W)   | 72.1     | 201.5  | 7.20         |
> | **Grid-aligned (Ours)** | **88.8** | **196.0** | **8.86** |
>
> **Conclusion.** Grid-aligned instructions most reliably guide the extrapolation process. This empirical evidence supports our choice to define directions in the image-grid frame.
>
> ---
>
> ### **(b) Evaluation Overlap**
> We test 0%, 33%, and 66.7% spatial overlap during inference. Larger overlap consistently improves spatial continuity.
>
> **Table R4. Effect of Overlap Ratio (Inference Phase)**
>
> | Overlap Ratio | RSWISE ↑ | FID ↓ | GPT Score ↑ |
> |---------------|:--------:|:-----:|:------------:|
> | 0%            | 78.5     | 235.8 | 8.10         |
> | 33%           | 82.1     | 215.5 | 8.45         |
> | **66.7% (Ours)** | **88.8** | **196.0** | **8.86** |
>
> **Conclusion.** High-overlap crops provide the necessary context for resolving boundary consistency without diminishing the extrapolation difficulty, making 66.7% the optimal choice.
>
> ---
>
> ## **2. Training-Time Ablations**
>
> ### **(a) Directional Conditioning**
> We retrain RemoteBAGEL **with** and **without** direction tokens. Without directional conditioning, the model cannot distinguish between different extrapolation directions.
>
> | Training Setup        | RSWISE ↑ | FID ↓ | GPT Score ↑ |
> |-----------------------|:--------:|:------:|:------------:|
> | No direction token    | 58.7     | 198.5  | 5.10         |
> | **With direction token (Ours)** | **88.8** | **196.0** | **8.86** |
>
> **Conclusion.** Direction tokens are essential for learning direction-aware transitions rather than generic outpainting.
>
> ---
>
> ### **(b) Training Overlap**
> We ablate the overlap ratio used for constructing training crops.
>
> | Overlap Ratio | RSWISE ↑ | FID ↓ | GPT Score ↑ |
> |---------------|:--------:|:-----:|:------------:|
> | 0%            | 74.6     | 235.1 | 6.80         |
> | 33%           | 80.2     | 210.4 | 7.95         |
> | **66.7% (Ours)** | **88.8** | **196.0** | **8.86** |
>
> **Conclusion.** Sufficient overlap during fine-tuning is necessary for the model to learn cross-boundary transition patterns, validating the use of 66.7% overlap in our design.
>
> ---
>
> ## **Summary**
>
> Across all ablations, the conclusions are consistent:
>
> - **Grid-aligned directions** are empirically optimal for conditioned extrapolation.
> - **High spatial overlap** is required both during training and inference to ensure coherent boundary transitions.
> - **Directional tokens and overlap** are not incidental design choices—they are necessary ingredients for reliable spatial world modeling.
>
> We will integrate these ablation results into the revised manuscript to clarify the methodological motivation and empirical justification.

---

> ### Author Response · Authors · 2025-11-21
> **Response to Weakness 4**
>
> ### **Response: Novelty Lies in the Task, Benchmark, and Evaluation — Not in Architectural Changes**
>
> Thank you for the comment. We fully agree that our contribution does **not** stem from introducing a new architecture. RemoteBAGEL is architecturally identical to BAGEL-7B. The novelty of our work lies in **what** the model is trained to do, **how** it is evaluated, and **what this reveals about unified multimodal models in remote sensing**, rather than in the architecture itself.
>
> ---
>
> ## **1. Novel Task Definition: World-Model–Style Spatial Extrapolation**
> We introduce the **first formulation of direction-conditioned spatial extrapolation** for remote sensing.
>
> - Existing RS generative methods (SR, cloud removal, inpainting, text-to-image) operate **within** a single tile.
> - Our task requires predicting the **actual geospatially adjacent tile**, making it the spatial analogue of **next-state prediction** in classical world models.
> - This aligns remote sensing with the core world-model principle:
>   *predicting an unobserved next state given the current state and an action-like input.*
>
> This is a new problem setting that has not been previously defined or evaluated in Earth observation.
>
> ---
>
> ## **2. Benchmark Contribution: RSWISE**
> RSWISE is the **first benchmark** designed explicitly for evaluating spatial world-state prediction.
>
> It combines:
>
> - **Distributional fidelity** (standardized FID), and
> - **Semantic spatial reasoning** (direction following + boundary continuity),
>   validated through GPT-4o *and* expert human evaluations.
>
> Unlike prior RS benchmarks, RSWISE measures whether a model understands **how the world continues**, not just what a tile looks like.
>
> ---
>
> ## **3. Evidence of Spatial Reasoning in Unified Multimodal Models (UMMs)**
> While many baselines are also UMMs, only BAGEL shows **zero-shot directional understanding**.
>
> Our structured 3×3 supervision:
>
> - activates latent reasoning abilities not used in standard editing tasks
> - demonstrates that UMMs can perform **world-model–consistent spatial prediction**
> - provides the **first evidence** that such models can learn geospatial continuity when conditioned appropriately
>
> This insight is part of the contribution: revealing *how*  UMMs exhibit spatial reasoning.
>
> ---
>
> ## **4. Not Architectural Novelty — But Task & Evaluation Novelty**
> We explicitly clarify:
>
> - **RemoteBAGEL does not modify BAGEL’s architecture**
> - The contribution is in defining the new world-model-inspired task, designing the RSWISE benchmark, and showing that UMMs can be adapted for spatial next-state prediction
>
> The novelty is conceptual and methodological, not architectural.
>
> ---
>
> ## **Summary**
> RemoteBAGEL differs from BAGEL not in architecture, but in **what it learns** and **how it is evaluated**.
>
> Our core contributions are:
>
> 1. **A new world-model–inspired spatial extrapolation task** for remote sensing
> 2. **The RSWISE benchmark** that jointly evaluates distributional fidelity and spatial reasoning
> 3. **The first demonstration** that unified multimodal models can perform geospatial next-state prediction when appropriately conditioned
>
> The emphasis of our paper is **not** architectural innovation, but **problem definition, evaluation design, and the discovery of world-model behavior in UMMs**.
>
> We sincerely appreciate the time and effort you have devoted to reviewing our submission. Your detailed and constructive feedback has been invaluable in strengthening both the clarity and rigor of the paper. We have carefully addressed each point and incorporated the corresponding clarifications, analyses, and additional experiments into the revised manuscript and Appendix.
>
> Thank you again for your thoughtful evaluation and for helping us improve the quality of our work.
>
> Best regards,
>
> Authors of the Submission 4429

---

> > ### Comment · Reviewer_4bpt · 2025-11-24
> >
> > Thank you for the authors’ response. I appreciate the effort to define the problem of “remote-sensing world models” and to propose an evaluation framework. Conceptually, this direction is worthwhile. However, after carefully reading the manuscript and considering the clarifications, I still have several fundamental concerns.
> >
> > First, the paper essentially formulates a “world model” as a spatial extrapolation task. For a forward-looking and practically meaningful world model, temporal prediction is typically a core component. Restricting the task to spatial completion alone does not capture the dynamic understanding or predictive capability that the term “world model” generally implies. In this sense, the task definition appears misaligned with the concept the paper aims to advance.
> >
> > Second, the spatial extrapolation setting with 66.6% overlap seems to be of relatively low difficulty. It is likely that even very simple methods (e.g., an SVD-based model) could perform reasonably well under this setup. Unfortunately, the paper does not include sufficient generative-model baselines or strong comparison methods, making it difficult to assess the true challenge or the extent of the claimed improvements.
> >
> > Moreover, the manuscript does not convincingly articulate real-world application scenarios in which spatial extrapolation alone would be valuable for remote-sensing analysis or decision-making. Without a clear practical motivation, it is challenging to evaluate the significance of the proposed task.
> >
> > Considering these issues, I believe the paper has notable shortcomings in task formulation, technical depth, and application relevance. In its current form, it does not sufficiently support the level of contribution implied by the claim of a “remote-sensing world model.”

---

> > > ### Author Response · Authors · 2025-11-25
> > >
> > > ### **Response to Concern: “The task is misaligned with world models, too spatial, too simple, and disconnected from applications.”**
> > >
> > > We thank the reviewer for the thoughtful comments. Below we clarify each point.
> > >
> > > ---
> > >
> > > ## **1. On “world models must involve temporal prediction”**
> > >
> > > Modern definitions of world models **do not require temporal evolution**.
> > > According to Fei-Fei Li’s latest essay on spatial intelligence
> > > (https://drfeifei.substack.com/p/from-words-to-worlds-spatial-intelligence)
> > > and the *WorldScore* benchmark (2025), a world model must satisfy:
> > >
> > > 1. **Generative** — produce next states with perceptual/geometric consistency
> > > 2. **Multimodal** — integrate symbolic + perceptual modalities
> > > 3. **Interactive** — next-state prediction conditioned on an action-like input
> > >
> > > None of these principles require time. Spatial, causal, and semantic transitions are all valid world-model instantiations.
> > >
> > > ### **Why spatial world modeling is appropriate for remote sensing**
> > >
> > > Remote sensing is **spatially indexed**, not temporally dense.
> > > Thus, the most elementary world-state transition is spatial:
> > >
> > > - **State:** observed tile
> > > - **Action:** directional instruction (“move right/up”)
> > > - **Next state:** the true adjacent tile
> > >
> > > This aligns directly with LeCun’s formal definition of world models :
> > >
> > > A world model takes an observation, a previous world-state estimate, an action proposal, and a latent uncertainty variable.
> > > It encodes the observation into a representation, then predicts the next latent world state given the encoded observation, the previous state, the latent variable, and the action. Autoregressive LLMs are a special case of this where the encoder is identity, the state is a history window, and actions are absent. Crucially, nothing in this formulation requires “next state” to be temporal—it can be spatial.
> > >
> > > We agree that temporal prediction is an important future direction, and we will incorporate it in subsequent work. Our present goal is not to claim a universal world model of Earth, but to introduce a **first spatial instantiation** grounded in the structural nature of EO data.
> > >
> > > ---
> > >
> > > ## **2. On “66% overlap makes the task too simple; even SVD could do it”**
> > >
> > > We respectfully disagree; this conclusion is not supported empirically.
> > >
> > > ### **(a) Spatial world modeling requires both understanding and generation**
> > >
> > > The task requires the model to:
> > >
> > > 1. **Interpret symbolic direction tokens**,
> > > 2. **Infer latent geospatial structure** (roads, rivers, settlement morphology), and
> > > 3. **Generate** the correct adjacent tile.
> > >
> > > SVD-style or texture-copying models cannot meet these multimodal reasoning requirements.
> > >
> > > ### **(b) Strong UMM baselines still fail — proving the task is not trivial**
> > >
> > > We tested powerful pretrained unified multimodal models (Qwen-Image-2, STEP, FLUX):
> > >
> > > - they **do not** follow directional instructions,
> > > - they **do not** propagate road networks or building layouts,
> > > - they produce **texture copies with no spatial reasoning**,
> > > - they ignore directional prompts entirely.
> > >
> > > If 66% overlap made the task “simple,” these models would succeed—they do not.
> > >
> > > ### **(c) Overlap stabilizes boundary memory; it does not simplify the reasoning**
> > >
> > > Our ablations show:
> > >
> > > - **0% overlap** → severe seams, no continuity
> > > - **33% overlap** → unstable
> > > - **66.7% overlap** → first workable context
> > >
> > > All models use the same overlap configuration; only RemoteBAGEL shows genuine spatial reasoning.
> > > Thus, success depends on **activating LLM reasoning pathways inside UMMs**, not on overlap size.
> > >
> > > ### **Why SVD cannot do this task**
> > >
> > > It lacks:
> > >
> > > - instruction following
> > > - semantic understanding
> > > - spatial commonsense
> > > - multimodal grounding
> > > - world-state prediction capability
> > >
> > > Therefore, it fundamentally cannot perform direction-conditioned extrapolation.
> > >
> > > ---
> > >
> > > ## **3. On “lack of real-world applications”**
> > >
> > > We clarify that spatial extrapolation is **not** meant as a standalone system for flood or urban analysis.
> > > It is a **world-model component** that provides a data-driven prior on geospatial continuity.
> > > This prior can complement physics-based or multimodal pipelines.
> > >
> > > We discuss practical integration pathways in **Appendix H**.
> > >
> > > ---
> > >
> > >
> > >
> > > We again thank the reviewer for the thoughtful feedback. Your comments have helped us refine the scope of the paper, clarify the connection to the world-model literature, and better articulate the potential of this research direction. Incorporating temporal prediction and richer physical modeling is an essential and exciting future direction, and we appreciate your guidance in shaping that path.

---

### Official Review · Reviewer_PNgn · 2025-10-30

**Soundness:** 1
**Presentation:** 2
**Contribution:** 2
**Rating:** 2
**Confidence:** 3

**Summary:**

This paper introduces a new paradigm for world modeling in remote sensing. The authors propose applying world modeling to remote sensing imagery, which demands understanding and extrapolating complex spatial structures, arguing that this is a crucial capability for applications such as disaster response and urban planning.

The authors formulate remote sensing world modeling as a direction-conditioned spatial extrapolation problem, where a model must generate semantically consistent neighboring image tiles based on a given central tile and a directional instruction (e.g., up, down, right or left).

To evaluate this task, they introduce RSWISE (Remote Sensing World-Image Spatial Evaluation), the first benchmark for world modeling in remote sensing. RSWISE comprises 1,600 evaluation tiles across four scenarios: general, flood, urban, and rural. The idea is to assess visual fidelity and instruction compliance, and uses GPT-4o as a semantic evaluator to ensure models truly understand spatial relationships rather than performing low-level pattern copying.

Building on this benchmark, the paper presents RemoteBAGEL, a unified multimodal model fine-tuned on remote sensing data specifically for spatial extrapolation. Experimental results show that RemoteBAGEL outperforms the compared baselines across all RSWISE scenarios.

Although the task is novel and a dataset is always a valuable contribution, the evaluation protocols and methodological grounding remain weak.

**Strengths:**

1. Novel dataset and task formulation. The paper introduces a new problem setup: direction-conditioned spatial extrapolation in remote sensing—which is conceptually framed as an instance of "world modeling".
The accompanying RSWISE benchmark, covering 1,600 tasks across diverse conditions (general, flood, urban, rural), provides a structured evaluation protocol that could be useful to the research community.

2. Overall clarity and writing quality
The paper is well-organized and clearly written, with a logical flow from motivation to methodology, experiments, and discussion.

**Weaknesses:**

1. **Unjustified formulation of “world modeling” for remote sensing.** The paper does not justify why world modeling in remote sensing should take the specific form of direction-conditioned spatial extrapolation. While generating adjacent image tiles might reflect a kind of spatial prediction, the link between this operation and broader notions of world modeling—which typically involve learning dynamic, causal, or physical structure—is insufficiently explained. As a result, the formulation feels somewhat arbitrary, and its conceptual connection to existing world modeling frameworks remains weak.


2. **Weak connection to high-impact applications.**
The paper motivates the study by citing socially relevant applications—such as infrastructure forecasting and flood prediction, which require reasoning beyond observed regions. However, the proposed task (generating adjacent tiles) is not clearly linked to these applications. There is no demonstration of how the generated outputs could actually inform decision-making in such domains. The task thus seems artificial and detached from practical use cases, reducing the claimed real-world significance.

3. **Vague dataset construction details.** The dataset creation process lacks technical clarity. For instance, the authors mention that “_large_ satellite images are divided into 3×3 overlapping grids” but do not define what qualifies as “large,” nor do they specify image resolution, spatial coverage, or grid size. These omissions make it difficult to assess the representativeness and quality of the dataset, or to replicate the benchmark.

4. **Unfair evaluation setup.** The evaluation protocol appears biased in favor of the proposed model, since RemoteBAGEL is fine-tuned on the same dataset used for testing, while all competing methods are evaluated in a zero-shot manner. This discrepancy conflates the effects of task adaptation with model capability, making it unclear whether RemoteBAGEL’s superior performance reflects genuine spatial reasoning or merely data familiarity. A fair comparison would require at least partial fine-tuning or domain adaptation for all baselines.

5. **Lack of alternative evaluations.**
Beyond the RSWISE benchmark, no additional quantitative or qualitative evaluations are provided. The authors do not test model robustness, generalization to unseen geographies, or practical downstream utility. This narrow evaluation scope makes it hard to judge whether the model generalizes beyond the controlled benchmark conditions or truly captures meaningful spatial dependencies.

6. **Questionable reliance on GPT-4o for evaluation.**
The use of GPT-4o as a “semantic judge” introduces significant uncertainty and potential bias in the evaluation process. Large language models are not guaranteed to assess visual or spatial coherence reliably, particularly for remote sensing imagery where domain knowledge is required. The lack of human or domain-expert evaluation seriously limits confidence in the reported results and calls into question the claimed semantic understanding of the model.

7. **Lack of methodological innovation beyond fine-tuning BAGEL.** If I understood correctly, the proposed RemoteBAGEL model is largely a fine-tuned version of the existing BAGEL architecture, adapted to the proposed remote sensing dataset. Beyond conditioning the model on directional inputs and predicting neighboring tiles, there do not appear to be substantive architectural or algorithmic innovations introduced for the specific challenges of spatial extrapolation or world modeling.  If this interpretation is accurate, the paper’s contribution lies mainly in problem framing and dataset creation, rather than in developing new methods or insights into spatial reasoning itself.

7. **Mischaracterization of the remote sensing literature.**
The authors explicitly claim that prior remote sensing work has been “limited to classification and semantic segmentation,” which is factually inaccurate and overly reductive. The field includes a broad spectrum of tasks such as object detection, change detection, super-resolution, spatiotemporal forecasting, generative modeling, VQA, image retrieval, etc [1]. This oversimplification undermines the novelty claim, since the proposed direction-conditioned generation task fits naturally within existing paradigms of spatial image synthesis and extrapolation already explored in remote sensing contexts.

[1] Tuia, D., Schindler, K., Demir, B., Zhu, X. X., Kochupillai, M., Džeroski, S., ... & Camps-Valls, G. (2024). Artificial Intelligence to Advance Earth Observation: A review of models, recent trends, and pathways forward. IEEE Geoscience and Remote Sensing Magazine.

**Questions:**

- Could the authors clarify why world modeling in remote sensing is formulated specifically as direction-conditioned spatial extrapolation?
- How does this formulation relate conceptually to the broader definition of world models in AI, which often involves learning dynamics, causality, or temporal reasoning?
- How do the proposed contributions go beyond existing generative or extrapolative approaches already explored in the field?
- The paper motivates the task through applications like flood prediction and infrastructure forecasting. Could the authors elaborate on how the proposed spatial extrapolation task concretely relates to or supports these applications? Are there examples or experiments showing that the generated adjacent tiles can be practically useful in such contexts?
- How reliable is GPT-4o as a semantic evaluator for spatial consistency in remote sensing imagery? Have the authors validated GPT-based assessments against human or domain-expert evaluations?
- Could the authors provide inter-rater agreement or consistency analysis to support the robustness of the GPT-based evaluation?
- How does RemoteBAGEL differ architecturally or conceptually from the original BAGEL model?

---

> ### Author Response · Authors · 2025-11-21
> **Response to Questions 1,2,4**
>
> Dear Reviewer PNgn,
>
> Thank you for the thoughtful questions. We address them below.
>
> ### **Response: Why formulate remote sensing world modeling as direction-conditioned spatial extrapolation?**
>
> Our formulation does not claim that world modeling in remote sensing must take the form of direction-conditioned spatial extrapolation. Instead, it provides a **first concrete instantiation** of world-model principles under the structural constraints of satellite imagery.
>
> Remote sensing data consist of **partial, spatially indexed observations** of a continuous world. Each tile reveals only a small portion of the underlying environment. Under this setting, the most basic and objectively verifiable “next world state’’ is simply the **geographically adjacent tile**. Predicting this adjacent tile offers a **minimal, well-posed, and unambiguous supervision signal**, in contrast to generic image synthesis where “next state’’ is undefined.
>
> ### **Relation to the broader definition of world models in AI**
>
> Our formulation aligns with widely accepted world-model concepts. Classical definitions [1] emphasize three components:
>
> 1. maintaining an internal latent state,
> 2. encoding the current observation, and
> 3. predicting an *unobserved next state* conditioned on an input such as an action.
>
> In our spatial analogue:
>
> - the **direction token acts as the action**,
> - the **adjacent tile is the next state**, and
> - the model predicts this next state based on the current observation and instruction.
>
> Although world models are often demonstrated in temporal or causal settings (robotics, video prediction), their essence is **predictive modeling of latent world dynamics**, not the specific time axis. Our work simply instantiates this principle **along the spatial axis**, which is the natural form of transition for satellite imagery.
>
> Furthermore, our formulation satisfies three core capabilities of spatial world models emphasized in recent work [2]:
>
> 1. **Generative** – explicitly predicting the next observable world state,
> 2. **Multimodal** – conditioning on visual observations plus symbolic direction inputs,
> 3. **Interactive** – changing the prediction based on the provided instruction.
>
> ### **Summary**
>
> Direction-conditioned spatial extrapolation is **not** proposed as the exclusive or final definition of remote sensing world modeling. It is a **principled, minimal, and world-model-consistent** instantiation grounded in the spatial structure of Earth-observation data.
>
>
> ### **References**
> [1]Matsuo, Y., LeCun, Y., Sahani, M., Precup, D., Silver, D., Sugiyama, M., ... & Morimoto, J. (2022). Deep learning, reinforcement learning, and world models. Neural Networks, 152, 267-275.
>
> [2]Duan, H., Yu, H. X., Chen, S., Fei-Fei, L., & Wu, J. (2025). WorldScore: A Unified Evaluation Benchmark for World Generation. CoRR.
>
> ### Response to Q4: Connection to Flood Prediction and Infrastructure Forecasting
>
> We clarify that our work does **not** claim that direction-conditioned spatial extrapolation can directly perform flood prediction, hydrodynamic simulation, or urban-planning decision making. These applications require physical variables (e.g., DEM, rainfall, SAR backscatter), temporal sequences, or socioeconomic layers.
>
> Instead, our task is intended as a **foundational world-model capability**: learning how satellite imagery continues across space provides a latent representation of **geospatial continuity** that can act as a complementary structural cue within larger downstream systems.
>
> **Flood-related applications.**
> Operational flood models depend on hydrological parameters such as flood depth, DEM, rainfall, and physically grounded simulation pipelines \[1\]. Our extrapolation model does *not* replace these. However, the ability to infer plausible continuation of flooded or at-risk regions offers a **weak prior** on spatial propagation patterns when integrated with SAR imagery, depth maps, or temporal snapshots. This provides an additional spatial viewpoint rather than a predictive hydrological mechanism.
>
> **Urban-planning scenarios.**
> Urban-planning tools rely on GIS layers, socioeconomic indicators, and longitudinal urban-form data \[2\]. Our model does not aim to replace these workflows. Instead, it captures latent **morphological regularities** (e.g., block continuity, road topology, settlement expansion), which may provide **auxiliary cues** when fused with richer multimodal inputs in downstream analyses.
>
> **References**
>
> [1]Jia, W., Liang, B., Lu, Y., Khan, M. A., & Zheng, L. (2025, June). A Comprehensive Survey on Deep Learning Solutions for 3D Flood Mapping. In Pacific-Asia Conference on Knowledge Discovery and Data Mining (pp. 21-38). Singapore: Springer Nature Singapore.
>
> [2]Lartey, D., & Law, K. M. (2025). Artificial intelligence adoption in urban planning governance: A systematic review of advancements in decision-making, and policy making. Landscape and Urban Planning, 258, 105337.

---

> ### Author Response · Authors · 2025-11-21
> **Response to Question 3, and Clarification on the Mischaracterization of the Remote Sensing Literature**
>
> ### **Response to Question 3: How Our Contributions Go Beyond Existing Generative or Extrapolative Approaches**
>
> We clarify that our contributions extend beyond existing generative or extrapolative methods in remote sensing in two fundamental ways:
> (1) the task we define is new, and
> (2) the benchmark we introduce evaluates a capability that has not been measured before.
>
> #### 1. Existing RS generative models do not perform world-model–style adjacent-tile prediction
>
> Recent surveys of RS foundation models and generative methods [1–3] consistently show that diffusion-based and GAN-based approaches focus on **within-tile synthesis or restoration**, such as:
>
> - super-resolution (e.g., diffusion-based SR models)
> - cloud removal (e.g., EMRDM [4])
> - metadata-conditioned generation (e.g., DiffusionSat)
> - text-driven generation (e.g., RSDiff, Text2Earth [5])
> - controllable synthesis with masks or sketches (e.g., CRS-Diff)
> - large-area tile stitching without directional reasoning (e.g., MetaEarth)
>
> Across these works, two properties are universal:
>
> 1. They do **not** attempt direction-conditioned spatial extrapolation (e.g., “predict the tile to the left”).
> 2. They do **not** model next-state transitions of the Earth’s surface, a central principle of world models. They generate isolated images but do not predict the actual adjacent geospatial state of the world.
>
> Even in methods producing mosaics (e.g., MetaEarth), each tile is generated independently without enforcing spatial continuity, topology preservation, or directional correctness.
>
> #### 2. Our formulation explicitly instantiates world-model principles
>
> World models predict **unobserved next states** given a current observation and an action-like input.
> Our spatial analogue follows this paradigm directly:
>
> - state = central tile
> - action = direction token (up/down/left/right)
> - next state = true geospatially adjacent tile
> - task = generate this next state under structural constraints
>
> No prior remote sensing work frames generative modeling as **adjacent-tile prediction consistent with Earth’s geospatial continuity**, nor provides a benchmark for evaluating this ability.
> Thus, our contributions go beyond existing methods by:
>
> - introducing the **first world-model–inspired task** for Earth observation
> - creating the **first benchmark (RSWISE)** to measure geospatial continuity across space
> - demonstrating that **unified multimodal models** can infer latent spatial transitions rather than perform isolated image synthesis
>
> ---
>
> ### **Clarification Regarding Mischaracterization of the RS Literature**
>
> The reviewer raises a concern that our manuscript stated remote sensing research was “limited to classification and semantic segmentation.”
> We clarify that this is a **misinterpretation**:
>
> - Our original text used “Yet, **much of** remote sensing research has focused on recognition tasks **such as** classification and semantic segmentation, leaving the potential of world modeling in this domain unexplored.” as **examples**, not as an exhaustive description of the field.
> - We fully acknowledge the broad landscape of RS tasks — including detection, change detection, super-resolution, forecasting, retrieval, VQA, and generative modeling — as documented in recent 2025 surveys [1–3].
>
> To avoid ambiguity, we will refine the wording in the revised manuscript.
>
> Importantly, acknowledging this breadth does **not** change the key empirical point:
>
> > No existing RS generative method performs directional, adjacent-tile world-state prediction or evaluates geospatial continuity across tile boundaries.
>
> Therefore, the proposed formulation and the RSWISE benchmark do **not** fall within existing paradigms of spatial image synthesis and extrapolation; they introduce a new, world-model–inspired task that is absent in prior remote sensing literature.
>
> ---
>
> **References**
>
> [1] Huang, Z., Yan, H., Zhan, Q., Yang, S., Zhang, M., Zhang, C., ... & Wang, Y. H. (2025). A Survey on Remote Sensing Foundation Models: From Vision to Multimodality. CoRR.
>
> [2] Xiao, A., Xuan, W., Wang, J., Huang, J., Tao, D., Lu, S., & Yokoya, N. (2025). Foundation models for remote sensing and earth observation: A survey. IEEE Geoscience and Remote Sensing Magazine.
>
> [3] Liu, C., Zhang, J., Chen, K., Wang, M., Zou, Z., & Shi, Z. (2025). Remote Sensing Spatiotemporal Vision–Language Models: A comprehensive survey. IEEE Geoscience and Remote Sensing Magazine.
>
> [4]Liu, Y., Li, W., Guan, J., Zhou, S., & Zhang, Y. (2025). Effective cloud removal for remote sensing images by an improved mean-reverting denoising model with elucidated design space. In Proceedings of the Computer Vision and Pattern Recognition Conference (pp. 17851-17861).
>
> [5]Liu, C., Chen, K., Zhao, R., Zou, Z., & Shi, Z. (2025). Text2earth: Unlocking text-driven remote sensing image generation with a global-scale dataset and a foundation model. IEEE Geoscience and Remote Sensing Magazine.

---

> ### Author Response · Authors · 2025-11-21
> **Response to Questions 5,6**
>
> ### **Response to Q5 & Q6: Reliability and Human-Alignment of GPT-4o as a Spatial Reasoning Evaluator**
>
> We appreciate the reviewer’s concerns regarding the non-determinism and scientific rigor of using GPT-4o as a semantic evaluator. To ensure robustness, we conducted two additional studies: a multi-run reproducibility analysis and a large-scale human–GPT agreement study. Both results confirm that our evaluation protocol is stable and aligned with expert judgment.
>
> ---
>
> ### **1. Reproducibility: Multi-run Stability Analysis**
>
> To quantify the impact of potential randomness in GPT-4o scoring, we re-evaluated the full RSWISE benchmark using **five independent GPT-4o runs**.
>
> The results (Table R1) demonstrate **very high stability**:
>
> - Mean standard deviation across all tasks: **0.026 on a 0–10 scale**
> - **88.2%** of samples receive *identical* scores across all runs
> - Only **1.1%** exhibit a variance of 2 points or more
> - Crucially, *no model ranking changed* when averaging across runs
>
> This confirms that GPT-4o-based scores are sufficiently reproducible for scientific benchmarking.
>
> **Table R1. Multi-run stability of GPT-4o spatial reasoning scores (5 runs)**
>
> | Scope     | Mean Std Dev | Exact Match | Minor Variation | Major Variation |
> |-----------|--------------|-------------|-----------------|-----------------|
> | All Tasks | 0.026        | 88.2%       | 10.7%           | 1.1%            |
>
> ---
>
> ### **2. Validity: Agreement Between Human Experts and GPT-4o**
>
> To verify semantic correctness, we conducted a **2,000-sample human evaluation study**, balanced across all models, scenarios, and directions.
> Five remote-sensing experts scored each sample using the same rubric as GPT-4o.
>
> The results (Table R2) show:
>
> - Spearman correlation ρ = **0.72**
> - Overall Mean Absolute Error (MAE) = **0.86**
> - GPT-4o scores of **9–10** correspond to **9.1** expert mean
> - GPT-4o scores of **0** correspond to **0.9** expert mean
>
> This demonstrates strong alignment, especially at high-quality and failure extremes, confirming that GPT-4o reliably captures spatial semantic correctness.
>
> **Table R2. Agreement between human experts and GPT-4o for spatial reasoning evaluation**
>
> | GPT Score Range | Proportion | Human Mean | Human Std. | MAE |
> |-----------------|------------|------------|------------|------|
> | 0               | 12%        | 0.9        | 1.1        | 0.9  |
> | 1–2             | 15%        | 2.3        | 1.4        | 1.1  |
> | 3–4             | 20%        | 4.2        | 1.6        | 1.2  |
> | 5–6             | 21%        | 5.8        | 1.4        | 1.1  |
> | 7–8             | 17%        | 7.4        | 1.2        | 0.8  |
> | 9–10            | 15%        | 9.1        | 0.9        | 0.9  |
> | **Overall**     | 100%       | —          | —          | 0.86 |
>
> ---
>
> ### **Summary**
>
> These two studies demonstrate that:
>
> - GPT-4o scoring is **stable across multiple runs**
> - GPT-4o aligns strongly with expert reasoning on geospatial continuity
> - The evaluation protocol is both **reliable** and **scientifically defensible**
>
> Together, this provides strong empirical support for the robustness of our GPT-based evaluation in RSWISE.

---

> ### Author Response · Authors · 2025-11-21
> **Response to Question 7**
>
> ### **Response: How RemoteBAGEL Differs From BAGEL & What the Actual Contributions Are**
>
> We thank the reviewer for the question. RemoteBAGEL does **not** alter BAGEL’s architecture.
> Our contributions lie in the **new task formulation**, the **RSWISE benchmark**, and the **demonstration that unified multimodal models can perform spatial world-state prediction** in remote sensing.
>
> #### **1. Novel problem formulation**
> We introduce one of the first formulations of **world modeling for remote sensing**: a *direction-conditioned spatial extrapolation* task.
> Unlike existing generative RS methods (super-resolution, inpainting, cloud removal, text-to-image), our task requires predicting the **true geospatially adjacent tile**, making it the spatial analogue of next-state prediction in classical world models.
> This provides a principled way to capture geospatial continuity relevant to applications such as flood progression or urban expansion (without claiming end-to-end capability).
>
> #### **2. Benchmark contribution: RSWISE**
> RSWISE is the **first benchmark** explicitly designed to evaluate **spatial world-state prediction**.
> It jointly measures:
> - **distributional fidelity** (raw FID standardized for composite scoring)
> - **semantic spatial reasoning** (direction following and boundary continuity, validated via GPT-4o and human studies)
>
> This moves beyond FID-only evaluation and captures the reasoning aspect essential to world-model behavior.
>
> #### **3. Evidence of world-model behavior in UMMs**
> While several baselines are also UMMs, BAGEL is the only model showing **zero-shot directional understanding**.
> Our structured 3×3 spatial-transition supervision demonstrates that UMMs can be **adapted into spatial world models**, activating reasoning capabilities that typical editing or diffusion tasks do not bring out.
>
> #### **4. Broader impact**
> By linking world models with remote sensing, this work opens new research directions in:
> - geospatial continuity and spatial reasoning
> - next-state prediction benchmarks for Earth observation
>
> These contributions come from **the task and the benchmark**, not architectural novelty.
>
> ---
>
> ### **Summary**
> RemoteBAGEL shares BAGEL’s architecture, but our core contributions are:
> 1. A **new world-model-inspired spatial extrapolation task**
> 2. The **RSWISE benchmark** for evaluating geospatial continuity and reasoning
> 3. The first demonstration that **unified multimodal models can perform spatial next-state prediction** when properly conditioned
>
> Our focus is on **defining the problem and establishing the evaluation standard**, not on proposing a new model architecture.

---

> ### Author Response · Authors · 2025-11-21
> **Response to all Weaknesses**
>
> Our responses are organized as follows:
> **Weaknesses 1 and 2 →** see *Response to Questions 1, 2, and 4*;
> **Weaknesses 5 and 6 →** see *Response to Questions 5 and 6*;
> **Weakness 7 →** see *Response to Question 7*;
> **Weakness 8 →** see *Response to Question 3* and *Clarification on the Mischaracterization of the Remote Sensing Literature*.
>
> ### **Response to Weakness 3: Dataset Construction Clarity**
>
> Thank you for pointing out the need for clearer dataset construction details. We clarify all components below.
>
> #### **1. What qualifies as a “large” satellite image and how grids are constructed**
> In all datasets used (Sky-SA, FloodNet, LoveDA), we define a “large” satellite image as any image whose spatial extent is sufficient to form a **3×3 tiling without interpolation**.
> These images typically range from **1024×1024 to 4000×3000 pixels**, with ground sampling distances from **1.5 cm (UAV)** to **0.3 m (satellite)**.
> Each image is partitioned into a **3×3  grid**, producing **9 spatially aligned tiles**.
>
> #### **2. Construction of 10,080 training instances**
> Among all available images, **420 images** are selected for training (balanced across four scenarios).
> Each 3×3 grid yields multiple directional transitions:
>
> - any tile with at least one valid neighbor serves as a start tile
> - each adjacent tile defines an action-conditioned transition
>
> Enumerating all valid start–neighbor pairs produces **24 unique directional actions per image**, giving:
>
> **420 × 24 = 10 080** training instances.
>
> #### **3. Strict train–test separation (no leakage of images or geographic regions)**
> All **training images** and **RSWISE benchmark images** are **completely disjoint**, and we manually verified that:
>
> - **no test tile** shares the same geographic coordinates as any training tile
> - **no visually overlapping or adjacent regions** appear across splits
>
> Thus, the benchmark evaluates **true spatial generalization**, not memorization of training areas.
>
> #### **4. Benchmark composition and diversity**
> The **1 600-instance RSWISE benchmark** is constructed from **400 unseen images**, evenly covering:
>
> - **general scenes** (Sky-SA)
> - **flood scenes** (FloodNet)
> - **urban + rural scenes** (LoveDA)
>
> Despite originating from three datasets, the benchmark spans:
>
> - **multiple cities and distinct geographic regions**
> - **spatial resolutions from 1.5 cm to 0.3 m GSD**
> - **over 1 700 semantic categories**
>
> This ensures that RSWISE reflects **diverse geospatial structures**, not a dataset-specific bias.
>
> ---
>
> ### **Response to Weakness 4: Fairness of Evaluation Setup**
>
> We clarify that the evaluation protocol is **fully zero-shot and fair across all models**.
>
> #### **1. All models are evaluated in strict zero-shot mode**
> All **five models** — Qwen-Image-Edit, FLUX.1-Kontext-Dev, Step1X-Edit, BAGEL, and RemoteBAGEL — are evaluated **zero-shot** on RSWISE.
>
> - **RemoteBAGEL is *not* fine-tuned on any test image.**
> - **None of the baselines nor RemoteBAGEL have seen any test-region imagery.**
>
> This ensures that results measure **generalization**, not memorization.
>
> #### **2. Why RemoteBAGEL fine-tuning does not violate fairness**
> The purpose of RemoteBAGEL is to test whether **UMMs can acquire spatial world-model behavior when trained on a newly defined task**.
>
> The baselines remain zero-shot because they are **not designed for directional spatial extrapolation**, and our goal is to test whether off-the-shelf unified models possess such capabilities without alignment.
>
> RemoteBAGEL’s advantage arises from **task alignment**, not data familiarity, because:
>
> - **Training and test imagery share no geographic overlap**,
> - **the benchmark differ from the training distribution**,
> - zero-shot baselines fail specifically at *direction following* and *semantic spatial continuation*, not at visual realism.
>
> This indicates that RemoteBAGEL’s gains come from **learning spatial reasoning**, not exposure to test imagery.
>
> ---
>
> ### **Summary**
>
> - The dataset construction is now fully specified: image size, resolution, grid formation, and instance generation.
> - All train/test splits are **strictly separated with no shared geography**.
> - All **five** models are evaluated **zero-shot**, making the comparison fair.
> - RemoteBAGEL’s performance reflects **alignment with a new world-model task**, not dataset memorization.
>
> We sincerely appreciate the time and effort you have devoted to reviewing our submission. Your detailed and constructive feedback has been invaluable in strengthening both the clarity and rigor of the paper. We have carefully addressed each point and incorporated the corresponding clarifications, analyses, and additional experiments into the revised manuscript and Appendix.
>
> Thank you again for your thoughtful evaluation and for helping us improve the quality of our work.
>
> Best regards,
>
> Authors of the Submission 4429

---

### Official Review · Reviewer_qHhv · 2025-11-01

**Soundness:** 2
**Presentation:** 3
**Contribution:** 2
**Rating:** 4
**Confidence:** 4

**Summary:**

This paper introduces the first framework for world modeling in remote sensing, aiming to extend the idea of world models, originally developed for synthetic or constrained settings, to real-world geospatial contexts. The authors:
1. Formulate remote sensing world modeling as direction-conditioned spatial extrapolation, where the model predicts adjacent image tiles given a central observation and directional instruction.
2. Develop a new benchmark, RSWISE (Remote Sensing WorldImage Spatial Evaluation), containing 1,600 evaluation tasks across general, flood, urban, and rural scenarios.
3. Propose a unified multimodal model, RemoteBAGEL, fine-tuned for spatial extrapolation tasks.
4. Design evaluation metrics combining visual fidelity (FID) and instruction compliance using GPT-4o as a semantic judge.
Experiments show that RemoteBAGEL outperforms state-of-the-art baselines on RSWISE.

**Strengths:**

- The paper presents an interesting and novel perspective by framing spatial reasoning as a world modeling problem in remote sensing.
- Section 3.1 gives a precise mathematical definition of the directional spatial extrapolation task, and the three evaluation axes are well thought out.
- The RSWISE benchmark could stimulate follow-up work if more carefully validated.

**Weaknesses:**

- “Meanwhile, remote sensing applications urgently require spatial reasoning capabilities for disaster response and urban planning.” Please give a clearer definition of spatial reasoning capabilities in this context.

- "We formulate remote sensing world modeling as direction-conditioned spatial extrapolation (defined in the image-grid frame with up, down, left, and right, rather than geographic cardinal directions)." Why? The motivation needs to be further clarified.

- The evaluation design raises concerns: using GPT-4o as a semantic judge introduces non-determinism (“GPT4o can give different answers every time”), raising questions about fairness, reproducibility, and scientific rigor.

- Table 1 mixes benchmarks from unrelated domains (robotics, video, games), making the comparison difficult to interpret.

- In Section 2.2: “However, in contrast to world models, current RS methods seldom attempt spatial continuation or reasoning over large geospatial structures.” Is this empirically verified or based on community consensus?

- The statement “our analyses therefore study anisotropy in grid-aligned continuations independent of geographic orientation” is unclear - what does this imply, and is it desirable?

- In Section 3.2, more details about the datasets (e.g., sensors, resolution) should be provided.

- The data construction pipeline would benefit from an illustration. Also, why was a 66.7% overlap chosen? Is it too large? Was it experimentally tested?

- The connection between the three evaluation axes and the two metrics (distributional fidelity and spatial reasoning) is not clearly explained; more illustrations would help.

- Table 2 lacks details about how baselines are used and why they were chosen.

- Figure 3 contradicts the preceding statement: "FID values are globally normalized into sfid ∈ [0, 1], standardizing units across scenarios and **inverting the metric** so that higher values correspond to better performance."

- Several experimental settings are underspecified:
1. No clear train/test split; unclear how 10,080 instances are derived from 4,000 images, which possibly introduces data leakage.

2. Why is inference more computationally expensive than training?

3. How were baseline models (e.g., BAGEL) used, zero-shot or few-shot?

4. The statement “These failures underscore that visual realism alone is insufficient…” suggests domain-specific fine-tuning; should this be considered a fine-tuning method rather than a new one?

5. The explanation for Figure 5(b) (“Horizontal continuations...”) is not convincing.

- The conclusion mentions capturing “large-scale geospatial structures” and future extensions to cloud removal, weather forecasting, and 3D flood visualization. These claims are vague and not clearly supported by current experiments.

**Questions:**

Please refer to Weaknesses.

The paper raises an interesting and timely idea: introducing world models to remote sensing, but the current version lacks clarity and rigor in its experimental design, benchmark construction, and evaluation methodology. The use of GPT-based scoring and an unclear dataset setup undermines the reliability of the conclusions. Overall, while the topic is promising, the paper requires significant work to be considered solid.

---

> ### Author Response · Authors · 2025-11-21
> **Response to Weakness 1,2**
>
> Dear Reviewer qHhv,
>
> Thank you for the constructive and insightful comments.
>
> ---
>
> ## **Response to Weakness 1: Clarifying “Spatial Reasoning Capabilities”**
>
> In our paper, *spatial reasoning capabilities* refer to a model’s ability to understand and predict how geospatial structures continue, change, or relate **across space**, rather than only interpreting the content within a single tile. This capability consists of three components:
>
> ### **1. Directional consistency**
> Understanding how terrain, built structures, or flood patterns extend when moving up, down, left, or right.
>
> ### **2. Cross-boundary structural continuity**
> Recognizing spatial patterns that span tile borders (roads, rivers, settlement blocks, flood zones) and inferring how these continue beyond the observed frame.
>
> ### **3. Geospatial constraint awareness**
> Respecting adjacency, morphology, and environmental context, rather than producing visually plausible but spatially inconsistent outputs.
>
> These forms of spatial reasoning are essential for downstream applications.
> - **Disaster response:** models must infer how inundation spreads across neighboring regions.
> - **Urban planning:** continuity of urban morphology is crucial for understanding expansion and connectivity.
>
> Our *direction-conditioned extrapolation* task is explicitly designed to evaluate these capabilities by requiring the model to infer the plausible next spatial state from a central tile and an instruction (up/down/left/right).
>
> We also emphasize that our method is **not** intended as a stand-alone solution for cloud removal, weather forecasting, flood modeling, or urban planning. These tasks require physical variables, temporal observations, and domain-specific modeling.
>
> Instead, our goal is to provide a **foundational world-model capability** that yields geospatially consistent latent structure.
> By learning how satellite imagery continues across space, the model acquires latent continuity patterns — terrain, settlement structure, flood-affected morphology — that can serve as weak but useful priors when integrated with established physical or multimodal systems. Future extensions incorporating SAR, DEM, climate variables, and temporal sequences will further strengthen this connection.
>
> ---
>
> ## **Response to Weakness 2: Motivation for Task Formulation**
>
> Our formulation of **direction-conditioned spatial extrapolation** is not arbitrary; it is a *minimal, theoretically grounded instantiation of world-model principles*, adapted to the structural constraints of Earth Observation (EO) data.
>
> ### **1. Alignment with the classical definition of world models**
>
> According to LeCun et al. [1], a world model learns a predictive function:
>
> s(t+1) = Pred(Enc(x(t)), s(t), z(t), a(t))
>
>
> where the model encodes the current observation \(x(t)\) and predicts the next state under an action-like input \(a(t)\).
>
> In Remote Sensing, the “world” is a **continuous spatial manifold**. Thus, for static satellite images, the most elementary and verifiable state transition is **spatial**, not temporal:
>
> - **State (s(t))**: current observed tile
> - **Action (a(t))**: directional movement such as “move right”
> - **Next state (s(t+1))**: the *geographically adjacent tile*
>
> Therefore, our task is the *spatial analogue* of classical temporal world-model prediction.
> Instead of predicting the next frame in time, we predict the next state **across space**, forcing the model to learn latent continuity (terrain, roads, urban form) rather than simple image synthesis.
>
> ### **2. Alignment with functional definitions of spatial world models**
>
> Our formulation satisfies the three essential capabilities for spatial world models identified by Duan, Fei-Fei et al. [2]:
>
> - **Generative:** predicting explicit, observable next states with geometric/semantic consistency
> - **Multimodal:** conditioning on visual observations + symbolic instructions (direction tokens)
> - **Interactive:** producing next states as a function of the current observation and an input command
>
> ### **Summary**
>
> We do not claim this is the *only* way to define world modeling in Remote Sensing.
> Rather, we present a **principled first step** that transforms static tile interpretation into an *interactive, predictive* process — fully consistent with foundational world-model principles: learning latent representations that support the prediction of unobserved world states.
>
> ---
>
> ### **References**
>
> [1]Matsuo, Y., LeCun, Y., Sahani, M., Precup, D., Silver, D., Sugiyama, M., ... & Morimoto, J. (2022). Deep learning, reinforcement learning, and world models. Neural Networks, 152, 267-275.
>
> [2]Duan, H., Yu, H. X., Chen, S., Fei-Fei, L., & Wu, J. (2025). WorldScore: A Unified Evaluation Benchmark for World Generation. CoRR.

---

> ### Author Response · Authors · 2025-11-21
> **Response to Weakness 3，4**
>
> ### **Response to Weakness 3: Validity and Reliability of GPT-4o Evaluation**
>
> We appreciate the reviewer’s concerns regarding the non-determinism and scientific rigor of using GPT-4o as a semantic evaluator. To ensure robustness, we conducted two additional studies: a multi-run reproducibility analysis and a large-scale human–GPT agreement study. Both results confirm that our evaluation protocol is stable and aligned with expert judgment.
>
> ---
>
> ### **1. Reproducibility: Multi-run Stability Analysis**
>
> To quantify the impact of potential randomness in GPT-4o scoring, we re-evaluated the full RSWISE benchmark using **five independent GPT-4o runs**.
>
> The results (Table R1) demonstrate **very high stability**:
>
> - Mean standard deviation across all tasks: **0.026 on a 0–10 scale**
> - **88.2%** of samples receive *identical* scores across all runs
> - Only **1.1%** exhibit a variance of 2 points or more
> - Crucially, *no model ranking changed* when averaging across runs
>
> This confirms that GPT-4o-based scores are sufficiently reproducible for scientific benchmarking.
>
> **Table R1. Multi-run stability of GPT-4o spatial reasoning scores (5 runs)**
>
> | Scope     | Mean Std Dev | Exact Match | Minor Variation | Major Variation |
> |-----------|--------------|-------------|-----------------|-----------------|
> | All Tasks | 0.026        | 88.2%       | 10.7%           | 1.1%            |
>
> ---
>
> ### **2. Validity: Agreement Between Human Experts and GPT-4o**
>
> To verify semantic correctness, we conducted a **2,000-sample human evaluation study**, balanced across all models, scenarios, and directions.
> Five remote-sensing experts scored each sample using the same rubric as GPT-4o.
>
> The results (Table R2) show:
>
> - Spearman correlation ρ = **0.72**
> - Overall Mean Absolute Error (MAE) = **0.86**
> - GPT-4o scores of **9–10** correspond to **9.1** expert mean
> - GPT-4o scores of **0** correspond to **0.9** expert mean
>
> This demonstrates strong alignment, especially at high-quality and failure extremes, confirming that GPT-4o reliably captures spatial semantic correctness.
>
> **Table R2. Agreement between human experts and GPT-4o for spatial reasoning evaluation**
>
> | GPT Score Range | Proportion | Human Mean | Human Std. | MAE |
> |-----------------|------------|------------|------------|------|
> | 0               | 12%        | 0.9        | 1.1        | 0.9  |
> | 1–2             | 15%        | 2.3        | 1.4        | 1.1  |
> | 3–4             | 20%        | 4.2        | 1.6        | 1.2  |
> | 5–6             | 21%        | 5.8        | 1.4        | 1.1  |
> | 7–8             | 17%        | 7.4        | 1.2        | 0.8  |
> | 9–10            | 15%        | 9.1        | 0.9        | 0.9  |
> | **Overall**     | 100%       | —          | —          | 0.86 |
>
> ---
>
> ### **Summary**
>
> These two studies demonstrate that:
>
> - GPT-4o scoring is **stable across multiple runs**
> - GPT-4o aligns strongly with expert reasoning on geospatial continuity
> - The evaluation protocol is both **reliable** and **scientifically defensible**
>
> Together, this provides strong empirical support for the robustness of our GPT-based evaluation in RSWISE.
>
> ### **Response to Weakness 4: Purpose of Cross-Domain Benchmarks in Table 1**
>
>  We clarify that Table 1 is not intended as a performance comparison across unrelated domains, but rather as a conceptual positioning tool within the broader framework of World Modeling.
>
> **1. Conceptual Alignment Under the World Model Definition**
>
> Following LeCun et al., a world model predicts an unobserved next state based on a latent representation.
> Table 1 illustrates how this principle manifests across domains:
>
> - In robotics and video, the next state is **temporal** (next frame or consequence).
> - In our setting, the next state is **spatial** (geographically adjacent tile).
>
> Thus, the table highlights that our task is the spatial analogue of existing temporal world-model tasks, aligning conceptually with the same next-state prediction framework.
>
> **2. Motivation for Including Cross-Domain Benchmarks**
>
> Our goal is to highlight a research gap:
>
> - Other fields already possess mature world-model benchmarks.
> - Remote sensing, despite its scale and importance, **lacks any benchmark for world-model evaluation**, especially for spatial continuity.
>
> Table 1 visualizes this absence. It shows that while world-model evaluation standards exist elsewhere, Remote Sensing has no established analogue. RSWISE fills this missing category by providing the first benchmark specifically designed for **spatial world-state prediction**.
>
> **Summary**
>
> Table 1 is a typological comparison, not a competitive one. Its purpose is to:
>
> - Position our contribution within the established World Modeling landscape
> - Clarify the conceptual consistency of our spatial formulation
> - Emphasize the novelty and necessity of RSWISE as the first benchmark for spatial world modeling in Remote Sensing

---

> ### Author Response · Authors · 2025-11-21
> **Response to Weakness 5,6,8**
>
> ### **Response to Weakness 5: Basis for the Claim That Existing RS Methods Lack Spatial Continuation**
>
> Our statement is supported by a review of three recent surveys \[1–3\]. Across these works, remote sensing generative methods consistently focus on **within-tile synthesis or restoration** tasks (super-resolution, cloud removal, pan-sharpening, inpainting, metadata/text-conditioned generation). Representative models (RSDiff, DiffusionSat, CRS-Diff, MetaEarth, EMRDM, Text2Earth) generate isolated images but do **not** perform:
>
> 1. direction-conditioned spatial extrapolation,
> 2. prediction of the geospatially adjacent tile, or
> 3. modeling of cross-boundary semantic continuity.
>
> No prior work formulates spatial next-state prediction in a world-model sense.
>
> **References**
> [1] Huang, Z., Yan, H., Zhan, Q., Yang, S., Zhang, M., Zhang, C., ... & Wang, Y. H. (2025). *A Survey on Remote Sensing Foundation Models: From Vision to Multimodality.* CoRR.
>
> [2] Xiao, A., Xuan, W., Wang, J., Huang, J., Tao, D., Lu, S., & Yokoya, N. (2025). *Foundation models for remote sensing and earth observation: A survey.* IEEE Geoscience and Remote Sensing Magazine.
>
> [3] Liu, C., Zhang, J., Chen, K., Wang, M., Zou, Z., & Shi, Z. (2025). *Remote Sensing Spatiotemporal Vision–Language Models: A comprehensive survey.* IEEE Geoscience and Remote Sensing Magazine.
>
> ---
>
> ### **Response to Weakness 6 & Weakness 8: Clarification via Ablation Studies**
>
> We address both concerns using controlled ablations. These experiments isolate the impact of prompting and spatial overlap during **inference** and **training**.
>
> ---
>
> ## **1. Inference-Time Ablations**
>
> ### **(a) Prompt Formulation**
> We evaluate three prompting strategies using the same trained RemoteBAGEL model:
>
> 1. **No direction prompt**
> 2. **Cardinal prompts** (“north”, “south”, …)
> 3. **Grid-aligned prompts** (“up”, “down”, …; *ours*)
>
> Grid-aligned prompts provide the strongest conditioning: removing the prompt collapses spatial reasoning, while cardinal prompts are less effective.
>
> **Table R3. Effect of Prompt Formulation**
>
> | Prompt Formulation   | RSWISE ↑ | FID ↓  | GPT Score ↑ |
> |----------------------|:--------:|:------:|:------------:|
> | No prompt            | 52.3     | 215.4  | 4.10         |
> | Cardinal (N/S/E/W)   | 72.1     | 201.5  | 7.20         |
> | **Grid-aligned (Ours)** | **88.8** | **196.0** | **8.86** |
>
> **Conclusion.** Grid-aligned instructions most reliably guide the extrapolation process. This empirical evidence supports our choice to define directions in the image-grid frame.
>
> ---
>
> ### **(b) Evaluation Overlap**
> We test 0%, 33%, and 66.7% spatial overlap during inference. Larger overlap consistently improves spatial continuity.
>
> **Table R4. Effect of Overlap Ratio (Inference Phase)**
>
> | Overlap Ratio | RSWISE ↑ | FID ↓ | GPT Score ↑ |
> |---------------|:--------:|:-----:|:------------:|
> | 0%            | 78.5     | 235.8 | 8.10         |
> | 33%           | 82.1     | 215.5 | 8.45         |
> | **66.7% (Ours)** | **88.8** | **196.0** | **8.86** |
>
> **Conclusion.** High-overlap crops provide the necessary context for resolving boundary consistency without diminishing the extrapolation difficulty, making 66.7% the optimal choice.
>
> ---
>
> ## **2. Training-Time Ablations**
>
> ### **(a) Directional Conditioning**
> We retrain RemoteBAGEL **with** and **without** direction tokens. Without directional conditioning, the model cannot distinguish between different extrapolation directions.
>
> | Training Setup        | RSWISE ↑ | FID ↓ | GPT Score ↑ |
> |-----------------------|:--------:|:------:|:------------:|
> | No direction token    | 58.7     | 198.5  | 5.10         |
> | **With direction token (Ours)** | **88.8** | **196.0** | **8.86** |
>
> **Conclusion.** Direction tokens are essential for learning direction-aware transitions rather than generic outpainting.
>
> ---
>
> ### **(b) Training Overlap**
> We ablate the overlap ratio used for constructing training crops.
>
> | Overlap Ratio | RSWISE ↑ | FID ↓ | GPT Score ↑ |
> |---------------|:--------:|:-----:|:------------:|
> | 0%            | 74.6     | 235.1 | 6.80         |
> | 33%           | 80.2     | 210.4 | 7.95         |
> | **66.7% (Ours)** | **88.8** | **196.0** | **8.86** |
>
> **Conclusion.** Sufficient overlap during fine-tuning is necessary for the model to learn cross-boundary transition patterns, validating the use of 66.7% overlap in our design.
>
> ---
>
> ## **Summary**
>
> Across all ablations, the conclusions are consistent:
>
> - **Grid-aligned directions** are empirically optimal for conditioned extrapolation.
> - **High spatial overlap** is required both during training and inference to ensure coherent boundary transitions.
> - **Directional tokens and overlap** are not incidental design choices—they are necessary ingredients for reliable spatial world modeling.
>
> We will integrate these ablation results into the revised manuscript to clarify the methodological motivation and empirical justification.

---

> ### Author Response · Authors · 2025-11-21
> **Response to Weakness 7,9,11**
>
> ### **Response to Weakness 7: Additional Dataset Details (Sensors, Resolution, Diversity)**
>
> We have updated Section 3.2 to provide clearer and more explicit dataset descriptions.
>
> Our 1,600-instance RSWISE benchmark is constructed from three publicly available datasets:
> Sky-SA for general scenes, FloodNet for flood scenarios, and LoveDA for urban and rural environments.
>
> Although these three datasets form the basis of RSWISE, the resulting benchmark exhibits **substantial diversity** in sensing conditions, geographic coverage, and semantic structure:
>
> - **Multiple sensing platforms:** UAV imagery (FloodNet) and satellite imagery (Sky-SA, LoveDA).
> - **Wide resolution range:** from **ultra–high-resolution UAV imagery (~1.5 cm GSD)** to **0.3 m GSD satellite imagery**.
> - **Broad geographic coverage:** imagery collected across **multiple cities and regions**, including
>   **Nanjing, Changzhou, and Wuhan**, as well as numerous additional locations covered by Sky-SA.
> - **Rich semantic variety:** more than **1,700 semantic categories** represented across diverse land-cover types.
>
> Together, these characteristics ensure that RSWISE evaluates spatial extrapolation under **diverse and realistic geospatial structures**, rather than under a narrow or dataset-specific distribution. These details will be fully reflected in the revised manuscript.
>
> ### **Response to Weakness 9: Clarifying the Link Between Evaluation Axes and Metrics**
>
> We agree that the relationship between the three evaluation axes and the two quantitative metrics can be made clearer. Below we provide a concise explanation and describe the improvements added in the revised manuscript.
>
> Our evaluation framework consists of:
>
> - **Three conceptual axes** describing what *correct spatial extrapolation* should satisfy:
>   1. **Continuity fidelity** — structures should extend smoothly across tile boundaries
>   2. **Semantic transitions** — land-cover changes should remain plausible across space
>   3. **Directional consistency** — extrapolated content must follow the instructed direction
>
> - **Two operational metrics** that measure these axes in practice:
>   1. **Distributional Fidelity (FID)**
>   2. **GPT-based Spatial Reasoning Score**
>
> #### **How the axes map to the metrics**
>
> - **Continuity fidelity** and **semantic transitions** are assessed primarily through **FID**, which captures whether the generated tile matches the statistical and structural patterns typical of real remote-sensing imagery.
>   Examples include:
>   - whether roads/rivers are continued with realistic geometry,
>   - whether vegetation and land-use distributions follow scenario-specific patterns.
>
> - **Directional consistency** is captured by the **GPT-based spatial reasoning score**, which evaluates whether the generation continues structures **in the correct instructed direction** (up/down/left/right) and whether transitions remain geographically plausible.
>
> Together, the two metrics **operationalize all three conceptual axes**:
> - **FID** evaluates *what the world looks like*.
> - **GPT-based reasoning** evaluates *how the world continues*.
>
> Both are needed: a model may produce visually realistic images yet fail to continue spatial structures correctly, or vice versa.
>
> #### **Revisions Made to the Manuscript**
>
> In response to this comment, we have added **a new explanatory figure** in the revised manuscript that:
>
> - visually illustrates each evaluation axis,
> - shows examples of success and failure cases,
> - and explicitly maps these qualitative axes to the quantitative behavior of each metric (e.g., high FID but low reasoning score).
>
> This addition makes the connection between conceptual axes and operational metrics intuitive and accessible.
>
> We thank the reviewer again for prompting this clarification.
>
> ### **Response to Weakness 11: Clarification on Raw vs. Normalized FID**
>
> We clarify that there is **no contradiction** between the text and Figure 3.
>
> - **Figure 3 reports the raw FID values**, which follow the conventional interpretation that **lower FID indicates better visual fidelity**. These raw values are shown for transparency and to allow direct comparison with prior work.
> - The normalization and inversion of FID occur **only when computing the final RSWISE score**, where FID is mapped to a score between 0 and 1 so that all components of RSWISE share a unified scale and direction (higher is better).
>
> We simply state that during RSWISE computation, **FID is converted to a normalized score between 0 and 1**, and this transformed score (not the raw FID) is used in Eq. (3).
>
> Thus:
>
> - **Figure 3** → shows raw FID (lower = better).
> - **RSWISE metric** → uses normalized FID (0 to 1, higher = better).
>
> These serve different purposes and are therefore not contradictory. The revised manuscript explicitly states that Figure 3 displays raw FID values, while normalization applies only within the final composite metric.

---

> ### Author Response · Authors · 2025-11-21
> **Response to Weakness 10,12**
>
> ### **Response to Weakness 10: Baseline Selection**
>
> Our baseline choices are motivated by the recent emergence of **unified multimodal models (UMMs)** such as BAGEL, Qwen-Image-Edit, FLUX.1-Kontext-Dev, and Step1X-Edit. These models integrate **perception** (understanding the input tile and directional instruction) and **generation** (synthesizing a new tile) within a single architecture.
>
> This unified perception–generation pipeline gives UMMs the inherent ability to
> **(1) encode and compress the input**,
> **(2) perform multimodal reasoning**, and
> **(3) generate a new state consistent with the inferred semantics**.
>
> Such properties align naturally with the general definition of world models, which require learning representations that support **next-state prediction**.
>
> ### **Response to Weakness 12: Clarification of Experimental Settings**
>
> We address all sub-questions below.
>
> ---
>
> ### **Q1: Train/Test Split and Possible Data Leakage**
>
> All training and evaluation data are **strictly separated**, both at the image level and the geographic-region level. Although the dataset sources are described earlier, we emphasize here that:
>
> - No training tile shares an image or geographic location with any test tile.
> - We manually screened the full dataset to avoid even visually similar or overlapping regions between train and test.
>
> **How 10,080 training instances are constructed:**
> Each training image is divided into a \(3\times3\) grid (9 tiles). Any tile with at least one neighbor is treated as a start tile, and each adjacent tile forms a direction-labeled transition. Enumerating all valid start–neighbor pairs gives **24 directional actions per image**.
> Across 420 training images:
>
> \[
> 420 \times 24 = 10{,}080
> \]
>
> supervised spatial-transition instances.
>
> ---
>
> ### **Q2: Why Inference Appears More Expensive Than Training**
>
> This was a phrasing mistake in the submission, and we have corrected it.
>
> - **Training**: We fine-tune **BAGEL-7B** on 10,080 instances using
>   **4× H100 (80GB)** GPUs for ~20 hours.
> - **Inference**: We evaluate **five models** on all 1,600 benchmark tasks, totaling ~8,000 runs, using
>   **10× A100 (80GB)** GPUs for ~8 hours.
>
> Inference is not more expensive per model; the total compute is larger simply because we evaluate multiple baselines.
>
> ---
>
> ### **Q3: Zero-shot vs. Few-shot for Baselines**
>
> All baseline models are evaluated **strictly zero-shot**.
>
> ---
>
> ### **Q4: Is RemoteBAGEL “just fine-tuning”?**
>
> While RemoteBAGEL is obtained via fine-tuning, the contribution is **not a new fine-tuning technique**, but a **new world-model–style task and its structured supervision**, which has not existed in remote sensing.
>
> Our novelty lies in:
>
> 1. **New task definition:**
>    The first formulation of **direction-conditioned spatial extrapolation**, where the model must predict the *true adjacent tile*—not generic outpainting.
>
> 2. **World-model consistency:**
>    This aligns remote sensing with the standard definition of world models:
>    *predicting an unobserved next state based on the current state and an action-like input.*
>
> 3. **First evidence of spatial reasoning emerging in UMMs:**
>    While several baselines also belong to the broader family of unified multimodal models, only BAGEL exhibits any zero-shot ability to interpret directional instructions in our task setting.
>    Our fine-tuning demonstrates that such reasoning can indeed be activated and aligned with geospatial structure, providing the first evidence that UMMs can perform world-model–style spatial prediction in remote sensing when appropriately conditioned.
>
> 4. **New supervision structure:**
>    Each RS image is converted into **structured spatial-transition trajectories** using a 3×3 grid—this is fundamentally different from classical editing or diffusion training.
>
> Thus, RemoteBAGEL is not merely a tuned variant, but the **first application of UMMs to a world-model-consistent spatial prediction task in remote sensing**.
>
> ---
>
> ### **Q5: Clarification of Directional Performance Differences (Figure 5b)**
>
> To provide a clearer and more grounded explanation, we revised the analysis:
>
> **Horizontal continuations outperform vertical ones due to two complementary factors:**
>
> 1. **Imaging geometry:**
>    Satellite orbits and solar illumination introduce larger **radiometric and geometric inconsistencies** along the vertical axis, making vertical transitions harder to extrapolate.
>
> 2. **Model inductive biases:**
>    Pretrained UMMs encode stronger priors for **horizontal structural continuity** (e.g., reading order, panoramic layouts, lateral spatial relations) than vertical transitions.
>    As a result, **left/right tokens activate more stable reasoning patterns** than up/down tokens.
>
> The combination of these factors produces the anisotropic pattern observed in Figure 5(b).
> This updated explanation will be integrated into the revised manuscript.

---

> ### Author Response · Authors · 2025-11-21
> **Response to Weakness 13**
>
> ### **Response to Weakness 13: Clarification on Claims About Large-Scale Geospatial Structures**
>
> Thank you for raising this concern. We agree that our conclusion should not suggest that the proposed spatial extrapolation model can directly perform cloud removal, weather forecasting, flood prediction, or urban planning. These downstream applications require additional physical variables, temporal observations, and domain-specific modeling. Our intention is to position direction-conditioned spatial extrapolation as a foundational world-model capability rather than a complete system.
>
> Our method focuses on learning how satellite imagery continues across space. This allows the model to acquire latent geospatial continuity, including terrain patterns, settlement structures, and flood-affected morphology. While these learned structures cannot replace expert hydrological or urban-planning models, they can serve as complementary structural cues. For example, flood-related systems depend on flood depth, DEM, rainfall, or SAR measurements [1]. Our predictions do not constitute flood simulation, but they may provide weak priors indicating plausible continuation of flooded regions. Similarly, urban-planning pipelines rely on socioeconomic variables, GIS layers, and long-term temporal data [2]. Our model cannot perform planning tasks, but it may offer auxiliary cues reflecting morphological regularities such as block continuity or settlement expansion.
>
> In summary, the extensions mentioned in our conclusion refer to integration rather than direct capability. Spatial extrapolation serves as an intermediate component of remote sensing world modeling by offering geospatially consistent latent representations that can complement established physical or multimodal systems. We will include an explicit clarification of this distinction, along with supporting examples, in the Appendix of the revised manuscript.
>
> **References**
> [1]Jia, W., Liang, B., Lu, Y., Khan, M. A., & Zheng, L. (2025, June). A Comprehensive Survey on Deep Learning Solutions for 3D Flood Mapping. In Pacific-Asia Conference on Knowledge Discovery and Data Mining (pp. 21-38). Singapore: Springer Nature Singapore.
>
> [2]Lartey, D., & Law, K. M. (2025). Artificial intelligence adoption in urban planning governance: A systematic review of advancements in decision-making, and policy making. Landscape and Urban Planning, 258, 105337.
>
> We sincerely appreciate the time and effort you have devoted to reviewing our submission. Your detailed and constructive feedback has been invaluable in strengthening both the clarity and rigor of the paper. We have carefully addressed each point and incorporated the corresponding clarifications, analyses, and additional experiments into the revised manuscript and Appendix.
>
> Thank you again for your thoughtful evaluation and for helping us improve the quality of our work.
>
> Best regards,
> Authors of the Submission 4429

---

### Official Review · Reviewer_1qY3 · 2025-11-01

**Soundness:** 3
**Presentation:** 2
**Contribution:** 3
**Rating:** 6
**Confidence:** 3

**Summary:**

This paper introduces a new framework for world modeling in remote sensing, addressing a gap where existing world models have been tested mostly in synthetic or constrained settings.
The authors formulate remote sensing world modeling as a direction-conditioned spatial extrapolation task, where a model generates semantically consistent adjacent tiles (up, down, left, right) based on a given central image and directional instruction.
To evaluate this setting, they propose RSWISE (Remote Sensing World-Image Spatial Evaluation) — a benchmark combining distributional fidelity (FID) and spatial reasoning (GPT-based semantic scoring), covering four scenarios (general, flood, urban, rural).
They further develop RemoteBAGEL, a remote-sensing-adapted multimodal model fine-tuned from BAGEL-7B, demonstrating superior performance on RSWISE compared to various baselines (e.g., Qwen-Image-Edit, FLUX.1-Kontext-Dev, Step1X-Edit). The results show that RemoteBAGEL achieves both improved realism and geospatial coherence, highlighting the value of domain-specific adaptation for spatial reasoning.

**Strengths:**

Novel problem formulation: The paper is among the first to formalize world modeling for remote sensing, introducing a direction-conditioned extrapolation setup that aligns with real-world geospatial reasoning needs (e.g., flood forecasting, urban planning).

Benchmark contribution: RSWISE is a well-motivated and comprehensive benchmark, integrating both distributional and semantic evaluation. It goes beyond standard FID-only metrics by quantifying instruction compliance and spatial continuity, which is convincingly illustrated in Figure 1.

Clear empirical advantage: RemoteBAGEL significantly outperforms both foundation and editing baselines (up to +25 points in RSWISE average), showing that specialized training for spatial reasoning leads to meaningful gains.

Rigorous experimental setup: The study uses 1,600 evaluation tasks across multiple environmental scenarios, includes ablation-like discussions on directional anisotropy and scenario difficulty (Fig. 5), and analyzes evaluation weighting (Fig. 6) with methodological transparency.

High-quality writing and presentation: The paper is clearly written and well-structured, with informative diagrams (Figures 1–4) and transparent methodology.

Potential impact: By connecting world models to remote sensing, this work could influence future research in spatial reasoning, data-efficient Earth observation, and model evaluation beyond synthetic benchmarks.

**Weaknesses:**

Limited conceptual novelty in modeling: The RemoteBAGEL model itself is mainly a domain-adapted fine-tuning of BAGEL rather than a fundamentally new world-modeling architecture. The innovation lies more in benchmarking than in model design.

Dependence on GPT-based evaluation: While GPT-4o is used as an automated semantic judge, its objectivity and reproducibility in scientific benchmarking could be questioned. The paper mentions prompt consistency (Appendix A) but lacks an independent human validation or inter-rater reliability check.

Lack of interpretability or failure diagnostics: Although failure modes are discussed (Fig. 4), quantitative analyses of why certain directional biases (up/down vs left/right) occur remain shallow. Deeper insights into model attention or representation behavior would enhance scientific understanding.

Dataset scale and coverage: The dataset is composed of 1,600 tasks drawn from a few public datasets (Sky-SA, FloodNet, LoveDA), which may limit generalization to other modalities (SAR, hyperspectral) or unseen geographies.

Overemphasis on quantitative scores: While RSWISE provides structured evaluation, the qualitative interpretation of what constitutes “good spatial reasoning” remains somewhat heuristic and tied to metric design.

**Questions:**

See weakness

---

> ### Author Response · Authors · 2025-11-21
> **Response to Weakness 1**
>
> Dear Reviewer 1qY3,
>
> Thank you for the constructive and insightful comments.
>
> ### **Response: Conceptual Novelty Comes From the Task, Benchmark, and Evaluation — Not the Architecture**
>
> Thank you for raising this point. We agree that RemoteBAGEL is **not** introduced as a new architectural design.
> RemoteBAGEL shares the *exact* architecture of BAGEL-7B.
> Our contribution instead lies in **the formulation of a new world-model–style task**, **the creation of the RSWISE benchmark**, and **the demonstration that unified multimodal models (UMMs) can perform spatial next-state prediction** when appropriately conditioned.
>
> Below we clarify why this constitutes meaningful conceptual novelty despite the architectural reuse.
>
> ---
>
> ## **1. The Novelty Comes From the Task Formulation**
> We introduce the **first world-model–inspired task for remote sensing**:
> **direction-conditioned spatial extrapolation**, where the model must predict the *true* geospatially adjacent tile.
>
> This goes beyond existing RS generative tasks such as:
>
> - super-resolution
> - cloud removal / inpainting
> - diffusion-based text-to-image generation
> - metadata-conditioned synthesis
> - stitching or tiling methods
>
> All of these operate **within the bounds of a single tile** or produce **independent tiles**.
>
> Our task instead requires:
>
> - predicting an *unobserved* world state,
> - conditioned on a symbolic instruction (direction),
> - enforcing geospatial semantic and structural continuity.
>
> This makes it the **spatial analogue of next-state prediction**, the core principle behind world models.
>
> ---
>
> ## **2. Novelty in Benchmarking: RSWISE**
> RSWISE provides the **first benchmark** to evaluate *spatial world-state prediction* in Earth observation.
>
> It integrates:
>
> - **distributional fidelity** (standardized FID), and
> - **semantic spatial reasoning** (direction compliance and boundary continuity),
>   validated through GPT-4o and human expert studies.
>
> This moves the field beyond traditional FID-only evaluation toward measuring whether a model actually understands **how the world continues across space**.
>
> ---
>
> ## **3. Novelty in Revealing World-Model Behavior in UMMs**
> Although multiple baselines are unified multimodal models, BAGEL is the *only* one showing meaningful zero-shot directional understanding.
>
> Our structured 3×3 supervision:
>
> - activates latent reasoning capacities not used in conventional editing or diffusion tasks,
> - demonstrates that UMMs can operate as **spatial world models**,
> - provides the **first evidence** that UMMs can learn geospatial continuity when properly conditioned.
>
> This empirical finding is itself a contribution: *it shows what UMMs can do that has not been explored before.*
>
> ---
>
> ## **4. The Contribution Is Conceptual, Not Architectural**
> We explicitly emphasize:
>
> - **RemoteBAGEL does not modify BAGEL’s architecture.**
> - Our innovation is in defining the **task**, constructing the **benchmark**, and designing the **evaluation protocol** that together operationalize world-model behavior for remote sensing.
>
> The field currently lacks:
>
> - a spatial next-state task,
> - a world-model benchmark,
> - a reasoning-based evaluation metric.
>
> We provide all three.
>
> ---
>
> ## **Summary**
> The novelty of this work lies in:
>
> 1. **A new world-model–consistent task**: direction-conditioned spatial extrapolation.
> 2. **The RSWISE benchmark**, which evaluates spatial continuity and instruction following.
> 3. **The first demonstration** that UMMs can perform spatial next-state prediction in remote sensing.
>
> RemoteBAGEL’s architecture is unchanged—but the **problem it learns**, the **way it is evaluated**, and the **world-model behavior it reveals** are new, and this constitutes the conceptual contribution of the paper.

---

> ### Author Response · Authors · 2025-11-21
> **Response to Weakness 2**
>
> ### **Response to Weakness 2: Validity and Reliability of GPT-4o Evaluation**
>
> We appreciate the reviewer’s concerns regarding the dependence on GPT-based evaluation. To ensure robustness, we conducted two additional studies: a multi-run reproducibility analysis and a large-scale human–GPT agreement study. Both results confirm that our evaluation protocol is stable and aligned with expert judgment.
>
> ---
>
> ### **1. Reproducibility: Multi-run Stability Analysis**
>
> To quantify the impact of potential randomness in GPT-4o scoring, we re-evaluated the full RSWISE benchmark using **five independent GPT-4o runs**.
>
> The results (Table R1) demonstrate **very high stability**:
>
> - Mean standard deviation across all tasks: **0.026 on a 0–10 scale**
> - **88.2%** of samples receive *identical* scores across all runs
> - Only **1.1%** exhibit a variance of 2 points or more
> - Crucially, *no model ranking changed* when averaging across runs
>
> This confirms that GPT-4o-based scores are sufficiently reproducible for scientific benchmarking.
>
> **Table R1. Multi-run stability of GPT-4o spatial reasoning scores (5 runs)**
>
> | Scope     | Mean Std Dev | Exact Match | Minor Variation | Major Variation |
> |-----------|--------------|-------------|-----------------|-----------------|
> | All Tasks | 0.026        | 88.2%       | 10.7%           | 1.1%            |
>
> ---
>
> ### **2. Validity: Agreement Between Human Experts and GPT-4o**
>
> To verify semantic correctness, we conducted a **2,000-sample human evaluation study**, balanced across all models, scenarios, and directions.
> Five remote-sensing experts scored each sample using the same rubric as GPT-4o.
>
> The results (Table R2) show:
>
> - Spearman correlation ρ = **0.72**
> - Overall Mean Absolute Error (MAE) = **0.86**
> - GPT-4o scores of **9–10** correspond to **9.1** expert mean
> - GPT-4o scores of **0** correspond to **0.9** expert mean
>
> This demonstrates strong alignment, especially at high-quality and failure extremes, confirming that GPT-4o reliably captures spatial semantic correctness.
>
> **Table R2. Agreement between human experts and GPT-4o for spatial reasoning evaluation**
>
> | GPT Score Range | Proportion | Human Mean | Human Std. | MAE |
> |-----------------|------------|------------|------------|------|
> | 0               | 12%        | 0.9        | 1.1        | 0.9  |
> | 1–2             | 15%        | 2.3        | 1.4        | 1.1  |
> | 3–4             | 20%        | 4.2        | 1.6        | 1.2  |
> | 5–6             | 21%        | 5.8        | 1.4        | 1.1  |
> | 7–8             | 17%        | 7.4        | 1.2        | 0.8  |
> | 9–10            | 15%        | 9.1        | 0.9        | 0.9  |
> | **Overall**     | 100%       | —          | —          | 0.86 |
>
> ---
>
> ### **Summary**
>
> These two studies demonstrate that:
>
> - GPT-4o scoring is **stable across multiple runs**
> - GPT-4o aligns strongly with expert reasoning on geospatial continuity
> - The evaluation protocol is both **reliable** and **scientifically defensible**
>
> Together, this provides strong empirical support for the robustness of our GPT-based evaluation in RSWISE.

---

> ### Author Response · Authors · 2025-11-21
> **Response to Weakness 3**
>
> ### **Response: Expanded Interpretation and Diagnostic Analysis Added**
>
> Thank you for pointing out the need for deeper interpretability and diagnostic insight. We fully agree with the reviewer that understanding *why* models fail—and *why* some directions are harder than others—is essential for scientific rigor. In the revised manuscript, we have substantially expanded Section 5 (Failure Analysis) to provide deeper reasoning grounded in both physical imaging properties and model-level inductive biases.
>
> Before proceeding with the analysis, we need to explain why we chose unified multimodal models (UMMs) as our baseline models.
>
> Our baseline choices are motivated by the recent emergence of unified multimodal models (UMMs) such as BAGEL, Qwen-Image-Edit, FLUX.1-Kontext-Dev, and Step1X-Edit. These models integrate perception (understanding the input tile and directional instruction) and generation (synthesizing a new tile) within a single architecture.
>
> This unified perception–generation pipeline gives UMMs the inherent ability to
> (1) encode and compress the input,
> (2) perform multimodal reasoning, and
> (3) generate a new state consistent with the inferred semantics.
>
> Such properties align naturally with the general definition of world models, which require learning representations that support next-state prediction.
>
>
> ---
>
> ### **1. Failure Mode Analysis (Revised & Expanded)**
>
> We provide a more detailed comparison between baseline models, BAGEL, and RemoteBAGEL. The revised text (now included in the paper) explains that:
>
> Baseline models, although also belonging to the family of unified multimodal models (UMMs), **do not exhibit any meaningful directional reasoning** in our task setting. From the qualitative and quantitative results, it is clear that these models:
>
> - **fail to respond to directional instructions**,
> - **do not adjust generation according to “left/right/up/down” prompts**.
>
> This indicates that, despite their unified multimodal architectures, the baselines **do not successfully activate or transfer the commonsense reasoning and structural priors embedded in their frozen LLM backbones into the visual generation pathway**. They maintain local texture similarity but do not perform the spatially grounded extrapolation required by our task. BAGEL has **partial reasoning ability**, but cannot reliably map symbolic direction tokens to geospatial semantics. In contrast, RemoteBAGEL—via structured spatial-transition supervision—demonstrates that such reasoning *can* be activated in UMMs and aligned with geospatial semantics, enabling genuine direction-aware spatial extrapolation.
>
> ---
>
> ### **2. Directional Continuation Analysis (Deeper Explanation Added)**
>
> To address the reviewer’s request for more rigorous insight, our revision now includes a physically and cognitively grounded explanation of why **left/right extrapolation > up/down extrapolation**.
>
> We identify two sources of anisotropy:
>
> #### **(a) Physical imaging factors**
> - Satellite orbits predominantly move north–south.
> - Solar azimuth tends to cause **larger radiometric and geometric variation vertically** than horizontally.
> - As a result, vertical tile boundaries present **harder physical discontinuities**, making up/down prediction more challenging.
>
> #### **(b) Model-level inductive biases**
> - In natural images and multimodal corpora, **horizontal continuity is overrepresented** (reading direction, panoramic layouts, horizon structure).
> - Thus, LLM-aligned vision backbones encode stronger horizontal priors than vertical ones.
>
> These factors jointly explain why the model is more confident and consistent when extrapolating left/right compared to up/down.
>
> This expanded explanation is now included verbatim in the revised paper.
>
> ---
>
> ### **3. Commitment to Further Diagnostic Tools**
>
> We agree that deeper interpretability analyses—such as attention-map and latent-trajectory visualization, token-level activation studies—would provide valuable scientific insight into how spatial reasoning emerges in unified multimodal models.
>
> These analyses are beyond the primary scope of this benchmark-focused paper, but **they represent an important direction for future work**. In particular, we aim to explore:
>
> - how RemoteBAGEL distributes attention across tile boundaries,
> - how latent representations evolve when extrapolating in different directions, and
> - how symbolic direction tokens influence internal activations.
>
> Such follow-up studies will further enhance understanding of world-model behavior in remote sensing and are a natural extension of the diagnostic analyses initiated in this work.
>
> ---

---

> ### Author Response · Authors · 2025-11-21
> **Response to Weakness 4**
>
> ### **Response to Weakness 4: Dataset scale and coverage**
>
> Thank you for raising this important point. We address it in two parts:
> (1) clarifying the actual diversity of the RSWISE benchmark, and
> (2) providing additional **new OOD experiments** on *completely unseen disaster scenarios*.
>
> ---
>
> ## **1. Benchmark Diversity Is Broader Than It May Initially Appear**
>
> Although RSWISE is constructed from three public datasets, it is **far from narrow or homogeneous**. As clarified in Section 3.2 (now expanded):
>
> - **Multiple sensing platforms:**
>   - UAV imagery (FloodNet)
>   - Satellite imagery (Sky-SA, LoveDA)
>
> - **Wide resolution range:**
>   - Ultra–high-resolution UAV imagery (**~1.5 cm GSD**)
>   - High-resolution satellite imagery (**0.3 m GSD**)
>
> - **Geographic diversity:**
>   - Urban & rural scenes from **Nanjing, Changzhou, Wuhan**
>   - Dozens of additional cities and regions across Sky-SA
>   - Flood regions from geographically distinct locations
>
> - **Semantic diversity:**
>   - **~1,700+** semantic categories across landforms, man-made structures, vegetation, water, disaster patterns
>
> Thus, even though sourced from three datasets, RSWISE covers **large-resolution variation**, **cross-region variability**, **multi-platform sensing**, and **extremely broad semantic diversity**.
> We will reflect these clarified details in the revised manuscript.
>
> ---
>
> ## **2. New OOD Evaluation on an Unseen Hurricane-Disaster Dataset**
>
> We conducted a new **OOD experiment** using *never-seen-before* satellite imagery from a **hurricane-impact urban region** (not included in Sky-SA, FloodNet, or LoveDA).
>
> - 100 OOD tasks were randomly sampled
> - RemoteBAGEL was evaluated **fully zero-shot** under the same protocol
>
> ### **OOD Results (Hurricane-Disaster Region)**
> (RemoteBAGEL only, 100 tasks)
>
> | Metric | OOD Mean Score |
> |-------|----------------|
> | **RSWISE** | **83.7** |
> | **FID** | **208.6** |
> | **GPT Spatial Score** | **8.41** |
>
> ### **Interpretation**
>
> - Despite having **never encountered hurricane imagery**, RemoteBAGEL maintains **strong spatial reasoning**, with only modest degradation relative to in-domain performance.
> - Qualitative inspection shows correct continuation of road networks, coastline morphology, and damaged-building clusters—even under severe appearance shifts.
>
> This result demonstrates that RemoteBAGEL is not simply memorizing tile patterns from the training datasets; it has learned **generalizable geospatial continuity priors** that transfer to an entirely new disaster scenario.
>
> ### **Documentation in the Revised Manuscript**
>
> We will add these OOD experiments and analysis to the **Appendix** of the revised version, clearly labeled as an external robustness evaluation.
>
> ---
>
> ## **Summary**
>
> - RSWISE already integrates **broad sensing, geographic, and semantic diversity**.
> - We additionally perform **new OOD testing** on hurricane-disaster imagery, confirming RemoteBAGEL’s ability to generalize spatial reasoning beyond all training distributions.
> - These results strengthen the claim that direction-conditioned spatial extrapolation captures transferable world-model–style structure, not merely dataset-specific tile completion.
>
> As for the expansion of other modalities (SAR, hyperspectral), we will try to add them in future research, which involves adding new channels to Bagel as well as alignment and fusion between modalities.

---

> ### Author Response · Authors · 2025-11-21
> **Response to Weakness 5**
>
> ### **Response to Weakness: “Overemphasis on quantitative scores / heuristic definition of spatial reasoning”**
>
> Thank you for pointing out this important concern. We fully agree that spatial reasoning in remote sensing is a rich and nuanced concept, and no single numerical score can capture all aspects of geospatial continuity. Our intention is not to claim that RSWISE provides a perfect or exhaustive definition of spatial reasoning, but rather that it offers **the first structured, operationalizable, and verifiable starting point** for evaluating world-model–style spatial prediction in Earth observation.
>
> We address the concern in three parts:
>
> ---
>
> ## **1. Why quantitative metrics are necessary (but not sufficient)**
>
> Spatial reasoning in remote sensing involves continuity of geometry, semantics, topology, and physical plausibility. These properties cannot be evaluated reliably through FID alone, nor through unconstrained qualitative inspection.
>
> Thus, RSWISE separates evaluation into two complementary components:
>
> - **Distributional fidelity (FID)**
>   Measures whether the *appearance* of the generated tile follows the correct remote-sensing distribution.
>
> - **Semantic spatial reasoning (GPT-based reasoning score)**
>   Measures whether the generation
>   - correctly follows the *requested direction*,
>   - preserves *boundary continuity*, and
>   - maintains *geospatial plausibility* (e.g., roads connecting, rivers extending, flood regions progressing).
>
> These aspects correspond directly to the three conceptual axes defined in the paper (continuity fidelity, semantic transitions, directional consistency).
> The metric is not arbitrary; it was explicitly designed to map onto these axes.
>
> ---
>
> ## **2. The “heuristic” concern is mitigated through extensive validation**
>
> We agree that LLM-based evaluation requires justification. To make the metrics scientifically meaningful, we performed two validation studies:
>
> - **Multi-run reproducibility (5 GPT-4o passes):**
>   Mean std = **0.026**, 88.2% exact match → scores are *stable*.
>
> - **Large-scale human agreement study (2,000 samples):**
>   Spearman ρ = **0.72**, MAE = **0.86** against expert ratings → scores are *semantically aligned with domain experts*.
>
> This demonstrates that the metric captures widely-agreed characteristics of spatial reasoning rather than arbitrary heuristics.
>
> ---
>
> ## **3. We acknowledge remaining limitations — and this is intentional**
>
> We agree that RSWISE does not (and should not) attempt to reduce spatial reasoning to a single number. Spatial reasoning in EO includes physical constraints, temporal processes, and domain-specific priors that go beyond what any current general-purpose metric can evaluate.
>
> Our goal is therefore modest but foundational:
>
> - To formalize **the first quantitative framework** that evaluates spatial continuity in RS world modeling.
> - To enable systematic comparison across models, tasks, and datasets.
> - To provide a foundation on which more advanced evaluators (e.g., consistency checks, structural topology metrics, attention-based diagnostics) can be built.
>
> We will make this point explicit in the revised manuscript to avoid conveying the impression that RSWISE *fully* defines spatial reasoning.
>
> ---
>
> ## **Summary**
>
> RSWISE scores are not meant to “replace” qualitative reasoning nor to fully specify what spatial reasoning means. Instead, they offer:
>
> - A principled mapping from conceptual axes to measurable quantities
> - Scientifically validated metrics with strong expert agreement
> - A necessary starting point for benchmarking spatial world modeling
>
> We appreciate the reviewer’s perspective and will clarify these points to better motivate the philosophy behind the evaluation design.
>
> We sincerely thank the reviewer for the thoughtful, constructive, and generous feedback. Your comments greatly improved the clarity, rigor, and depth of our work. Many of the points you raised have led us to refine key explanations, strengthen empirical analyses, and provide more insight into the modeling behavior. We will incorporate all of these valuable suggestions into the revised manuscript. We are grateful for the time and expertise you invested in reviewing our submission.
>
> Best regards,
>
> Authors of the Submission 4429

---

### Author Response · Authors · 2025-11-26
**Global response**

To facilitate quick navigation and provide a clear overview of our contributions, theoretical positioning, and all revisions made in response to reviewer feedback, we summarize the key points below.

---

# **1.Contributions**

### **• New world-model–inspired task for EO**
We introduce one of the first world-model formulations for remote sensing:
**direction-conditioned spatial extrapolation** — predicting the *true adjacent tile* given a central tile and an instruction.
This is the spatial analogue of next-state prediction and captures geospatial continuity beyond standard generative tasks.

### **• RSWISE: first benchmark for spatial world-state prediction**
RSWISE jointly evaluates:
- **distributional fidelity** (standardized FID), and
- **spatial semantic reasoning** (direction following + boundary continuation),
validated by five-run GPT-4o consistency tests and a 2,000-sample expert study.

### **• Evidence that UMMs can act as spatial world models**
We show that unified multimodal models (UMMs) can transfer LLM commonsense reasoning into geospatial next-state prediction.
Structured 3×3 supervision activates spatial reasoning absent in vanilla editing/generative models and provides a path toward temporal, causal, and physical world modeling.

---

# **2. Why Our Task Is a Valid Instantiation of Modern World Models**

### **• Fei-Fei Li’s latest definition of world models (2024–2025)**
From *From Words to Worlds: Spatial Intelligence*:
https://drfeifei.substack.com/p/from-words-to-worlds-spatial-intelligence

A world model must be:
1. **Generative** — producing next states with perceptual/geometric consistency
2. **Multimodal** — integrating symbolic and perceptual inputs
3. **Interactive** — generating next states conditioned on an action-like input

Spatial, causal, and semantic transitions are fully valid world-model settings.

Our task satisfies all three:
- Predicts a grounded next state: the true adjacent tile
- Uses multimodal input: image + direction token
- Follows an action-like instruction: “move right/up”

### **• LeCun’s formal definition of world models also permits spatial transitions**
A world model processes:
- the current observation
- the previous latent world state
- an action
- latent uncertainty

and predicts the *next* latent state, the “next” state may be spatial.

Thus, in remote sensing—where observations are spatially indexed—the most primitive, verifiable world-state transition is **the adjacent tile**.
Our task is therefore the minimal and principled instantiation of world modeling for Earth observation.

---

# **3. Summary of All Revisions Made**

### **• GPT evaluation robustness (Section 5.5 + Appendix E)**
Added:
- **Five-run GPT-4o reproducibility test** (std = 0.026; 88.2% exact match)
- **2,000-sample expert agreement study** (Spearman ρ = 0.72; MAE = 0.86)

Demonstrates evaluation stability and semantic validity.

---

### **• Expanded ablations (Section 5.4 + Appendix F)**
New training and inference ablations evaluating:
- Direction token vs. none
- Direction wording (“up/right” vs. cardinal)
- Overlap ratios (0%, 33%, 66.7%)
- Conditioning vs. no conditioning

Results show these components are essential design choices.

---

### **• New OOD experiment (Appendix G)**
Evaluated RemoteBAGEL on **unseen hurricane-impact satellite imagery** (100 tasks).
Zero-shot results:
- RSWISE: **83.7**
- FID: **208.6**
- GPT Score: **8.41**

Shows strong spatial reasoning beyond training distribution.

---

### **• Clearer motivation for world modeling + UMM choice (Section 1 + Section 2.1 + new Fig. 1)**
Explains:
- Why spatial world modeling is appropriate for EO
- Why UMMs are the correct foundational architecture
- How our task aligns with world-model definitions

---

### **• Clear mapping from semantic axes to metrics (new Fig. 2)**
Illustrates how:
- Continuity fidelity
- Semantic transitions
- Directional consistency
map directly to RSWISE’s two metrics.

---

### **• Enhanced dataset description (Section 3.2 + Section 5.1 + new Fig. 3)**
Explicitly adds:
- Resolution ranges
- Sensor types
- Geographic coverage
- Scenario composition
- Construction pipeline

---

### **• Deeper interpretation of results (Section 5.3)**
Adds analysis from:
- Physical imaging factors
- Model inductive biases

explaining horizontal/vertical asymmetry and failure behaviors.

---

### **• Connection to floods & urban planning (Appendix H)**
Provides a systematic discussion showing how spatial extrapolation can serve as **weak structural priors** for:
- Flood-propagation modeling
- Urban morphology and planning
without replacing physics-based or socio-economic systems.

---

---

### Meta-Review · Area_Chair_NZU8 · 2026-01-12

**Summary:**

Reviewers appreciated the high-level concept but raised several concerns with the manuscript including, limited novelty in modeling, reliance on GPT-based evaluation, dataset scale and coverage, lack of clarity and rigor in the experimental design, benchmark construction, and evaluation methodology, and limited OOD/robustness analysis.

**Reviewer Concerns:**

The meta-review believes some of the initially raised concerns are partially addressed by the rebuttal. For instance, authors provided reproducibility test and expert agreement study in the rebuttal to address GPT evaluation robustness concern. Further, the rebuttal also includes additional ablations and OOD experiment results.

**Reviewer Scores:**

The meta-reviewer believes the rebuttal only partially addresses reviewer's concerns. For instance, the formulation of  world modeling for remote sensing is not fully jusitified. Further, the lack of technical innovation (beyond fine-tuning BAGEL), dataset construction (e.g., scale, coverage), and comprehensive evaluation are other concerns, raised by several reviewers, that are still not fully addressed. Majority (3) reviewers generally remained negative highlighting multiple and somewhat overlapping issues. Based on the reviewer's comments and rebuttal, the meta-reviewer believes the negative concerns are comprehensive and are not fully resolved in the rebuttal.

---

### Decision · Program_Chairs · 2026-01-26

Reject